# Reaction dynamics of P($^4$S) + O$_2$(X $^3\Sigma^-$) → O($^3$P) + PO(X $^2\Pi$) on a global CHIPR potential energy surface of PO$_2$(X $^2$A$_1$): implication for atmospheric modelling

Guangan Chen[1], Zhi Qin*[12], Ximing Li[1], Linhua Liu*[123]

[1]School of Energy and Power Engineering, Shandong University, 250061, Jinan, China.

[2]Optics and Thermal Radiation Research Center, Institute of Frontier and Interdisciplinary Science, Shandong University, 266237, Qingdao, China

[3]School of Energy Science and Engineering, Harbin Institute of Technology, 150001, Harbin, China

*Correspondence to*: Zhi Qin (z.qin@sdu.edu.cn) and Linhua Liu (liulinhua@sdu.edu.cn)

**Abstract.** Reaction dynamics of P($^4$S) + O$_2$(X $^3\Sigma^-$) → O($^3$P) + PO(X $^2\Pi$) is thought to be important in atmospheric and interstellar chemistry. Based on the state-of-the-art *ab initio* energy points, we analytically construct a global potential energy surface (PES) for the ground state PO$_2$(X $^2$A$_1$) using the combined-hyperbolic-inverse-power-representation (CHIPR) method. A total of 6471 energy points are computed by the multireference configuration interaction method with the Davidson correction and aug-cc-pV5Z basis set. The analytical CHIPR PES reproduces *ab initio* energies accurately with a root-mean-square deviation of 91.5 cm$^{-1}$ (or 0.262 kcal/mol). The strongly-bound valence region of the PES has complicated topographical features with multiple potential wells and barriers. The attributes of the important intermediates are carefully validated with our geometry optimization results, previous experimental and computational results. Finally, the reaction probability, integral cross sections and rate constants for P($^4$S) + O$_2$(X $^3\Sigma^-$) → O($^3$P) + PO(X $^2\Pi$) are calculated using the quasi-classical trajectory and time-dependent wave packet methods. The trends of probability and integral cross section versus the collision energy can be divided into three stages, which are governed by the entrance barriers or exothermicity of the reaction. The rate constant demonstrates strong Arrhenius linear behavior at relatively low temperatures, but it deviates from this pattern at high temperatures. The calculated cross sections and rate constants are helpful for modelling the phosphorus chemistry in atmosphere and interstellar media.

## 1 Introduction

Phosphorus (P) is one of the essential biogenic elements found in all known organisms. Its compounds are abundant in the biological systems and greatly contribute to the basic biological functions (Maciá, 2005). Much of the P on Earth's surface is locked up in mineral phosphate forms with fairly poor bioavailability, while the P compounds with low oxidation are generally more reactive and accessible for potential prebiotic chemistry (Todd, 2022) but unstable under terrestrial redox conditions (Pasek, 2008).

One source of the P compounds with low oxidation on Earth's surface is interstellar dust particles (IDPs), which account for 99% of the total amount of incoming extra-terrestrial material each year (Plane et al., 2018), containing P with about 8% abundance (Lodders, 2003). The ablation of IDPs in the upper atmosphere of terrestrial planets delivers PO and P (Carrillo-Sánchez et al., 2020), a part of which might undergo a series of chemical processes to form bioavailable H$_3$PO$_3$ before reaching the Earth's surface. The corresponding P chemistry networks (Douglas et al., 2019; Douglas et al., 2020; Plane et al., 2021) were predicted by the electronic structure theory, given by

$$P(^4S) + O_2 \rightarrow O + PO \qquad \qquad \text{R1}$$
$$PO + O_2 \rightarrow O + PO_2 \qquad \qquad \text{R2}$$
$$PO_2 + H(+M) \rightarrow HPO_2 \qquad \qquad \text{R3}$$
$$HPO_2 + H_2O(+M) \rightarrow H_3PO_3 \qquad \qquad \text{R4}$$

It suggests that the meteor-ablated P is likely oxidized by the reaction R1 first. Hence, the dynamic study for the reaction R1

will contribute to a deeper understanding of the chemical evolution of P and PO in the Earth's atmosphere.

The reaction R1 is also important in astrochemistry. For example, PO has been widely observed in the interstellar medium (ISM) (Tenenbaum et al., 2007; De Beck et al., 2013; Ziurys et al., 2018; Lefloch et al., 2016; Rivilla et al., 2020) and is considered to be the main reservoir for gas-phase P in the ISM (Ziurys et al., 2018; Rivilla et al., 2020). There is some evidence to suggest that PO was present in cometary ices before the birth of the sun (Rivilla et al., 2020) and the comets possibly provided a major source of prebiotic phosphorus compounds (Maciá et al., 1997). Hence, the investigation on the formation of PO is helpful for modelling its abundance in the ISM.

Given its important role in the interstellar and atmospheric chemistry, the reaction R1 was investigated in several experimental and theoretical works (Douglas et al., 2019; Husain and Norris, 1977; Husain and Slater, 1978; Clyne and Ono, 1982; Henshaw et al., 1987; Gomes et al., 2022). Previous experiments presented its rate constants at a specified temperature about 300 K (Husain and Norris, 1977; Husain and Slater, 1978; Clyne and Ono, 1982; Henshaw et al., 1987) and their values diverged by an order of magnitude. A recent experiment has determined the rate constants of $P(^4S) + O_2 \rightarrow O + PO$ at temperatures ranging from ~ 200 to 750 K (Douglas et al., 2019). In the same work, the rate constants of $P(^4S) + O_2(X\ ^3\Sigma^-) \rightarrow O(^3P) + PO(X\ ^2\Pi)$ were also computed using the Rice-Ramsperger-Kassel-Markus (RRKM) theory with the molecular geometries optimization at the B3LYP/aug-cc-pVQZ (AVQZ) level. Subsequently, the state-of-the-art *ab initio* method was used to predict the minimum energy path (MEP) of this reaction (Gomes et al., 2022) and the rate constants were computed employing the standard transition state theory (TST). These two theoretical rate constants reproduced the experimental results to some extent and the difference between them came mainly from the predicted barrier heights. The limitation of the former is that the B3LYP method is not good at predicting the barrier heights (Zhao and Truhlar, 2008; Peverati and Truhlar, 2012). Also, the statistical RRKM and TST theories may fail due to unincluded non-statistical dynamic effects (Carpenter, 1998; Thomas et al., 2008), so a dynamic study for this reaction is required. Moreover, their analytical forms of the rate constants were obtained by fitting predictive data lower than 1000 K, so the fitted rate constants at high temperatures may not be accurate. Accurately modelling the rate constants of $P(^4S) + O_2(X\ ^3\Sigma^-) \rightarrow O(^3P) + PO(X\ ^2\Pi)$ in a wide temperature range is desired, because the temperatures of ablation IDPs could reach more than 2500 K.

The reactants $P(^4S) + O_2(X\ ^3\Sigma^-)$ and products $O(^3P) + PO(X\ ^2\Pi)$ are connected to the lowest doublet and quartet states of $PO_2$, in which the doublet state $(X\ ^2A_1)$ plays a major role in this reaction (Gomes et al., 2022). The molecular geometries and vibration frequencies of $PO_2(X\ ^2A_1)$ have been well studied by several theoretical (Lohr, 1984; Kabbadj and Liévin, 1989; Jarrett-Sprague and Hillier, 1990; Cai et al., 1996; Francisco, 2002; Xianyi et al., 2008; Liang et al., 2013; Xu et al., 1996; Bauschlicher, 1999) and experimental (Cordes and Witschel, 1965; Davies and Thrush, 1968; Drowart et al., 1972; Verma and McCarthy, 1983; Kawaguchi et al., 1985; Hamilton, 1987; Qian et al., 1995; Lei et al., 2001; Lawson et al., 2011) methods. The recent measurement reported $R_{PO} = 2.771\ a_0$ and $\theta_{OPO} = 135.3°$ with the $C_{2v}$ symmetry (Kawaguchi et al., 1985), along with the vibration frequencies of 1075.4 cm$^{-1}$, 397.3 cm$^{-1}$ and 1327.54 cm$^{-1}$ for the symmetrical stretching $(\omega_1)$ (Lei et al., 2001), bending $(\omega_2)$ (Lei et al., 2001) and antisymmetric stretching $(\omega_3)$ (Lawson et al., 2011), respectively. It is worth noting that the potential energy surface (PES) can yield physical insight into the reaction path, energy transfer and structure of intermediates. The analytical form of a global PES modeled by *ab initio* energies can accurately predict the barrier height and it is the first step toward molecular simulations, such as the reactive or non-reactive of collisions and photodissociation within the system (Conway et al., 2020; Caridade et al., 2013; Schmidt et al., 2013). Therefore, the dynamic study carried out on such a PES is reliable and the information including the reaction probability and integral cross section (ICS) can be well predicted. The first analytical PES of $PO_2(X\ ^2A_1)$ was constructed using the B3P86/6-311++G(3df, 3pd) energy points (Zeng and Zhao, 2012), but the dissociation energy was well beyond the experimental value and the intermediates were not all predicted. For performing a high-precision dynamic study of the title reaction, it is necessary to develop a global PES of $PO_2(X\ ^2A_1)$, in which the potential energies should be calculated at the state-of-the-art *ab initio* method and the reaction path should be well reproduced and carefully validated.

This work aims to establish a global PES for the ground state $PO_2(X\ ^2A_1)$, so as to present the dynamic study on $P(^4S) + O_2(X\ ^3\Sigma^-) \rightarrow O(^3P) + PO(X\ ^2\Pi)$. The state-of-the-art *ab initio* method was applied to calculate the potential energies of $PO_2(X\ ^2A_1)$.

The analytical PES was then generated using the combined-hyperbolic-inverse-power-representation (CHIPR) method (Varandas, 2013; Rocha and Varandas, 2020, 2021). The rate constants for the $P(^4S) + O_2(X\,^3\Sigma^-) \rightarrow O(^3P) + PO(X\,^2\Pi)$ reaction at temperatures ranging below 5000 K were obtained using the quasi-classical trajectory (QCT) (Peslherbe et al., 1999; Li et al., 2014) and time-dependent wave packet (TDWP) (Zhang and Zhang, 1993, 1994) methods, and were compared with available experimental and theoretical results. Moreover, the state-specified reaction probability and ICSs were provided in order to get a deeper understanding of this reaction.

**2 *Ab initio* calculations**

All *ab initio* calculations of the ground state $PO_2(X\,^2A_1)$ were carried out using the MOLPRO 2015 software package (Werner et al., 2020; Eckert et al., 1997) with the $C_s$ (A′) symmetry point group. The Dunning-type aug-cc-pV5Z (AV5Z) basis set (Dunning et al., 2001; Martin and Uzan, 1998; Woon and Dunning Jr, 1993) was applied. The calculation processes are as follows. Firstly, the Hartree-Fock (HF) method was used to obtain the single-configuration wavefunction of $PO_2(X\,^2A_1)$. The relevant multi-configuration wavefunctions were generated by the full-valence complete active space self-consistent field (CASSCF) method (Knowles and Werner, 1985) based on the HF wavefunction. Finally, the dynamic correlation energies were considered by the internally contracted multireference configuration-interaction method including the Davidson correction [MRCI(Q)] (Knowles and Werner, 1988; Werner and Knowles, 1988), in which the CASSCF wavefunctions were set as reference. In the CASSCF and MRCI(Q) calculations, 12 active molecular orbitals (9A′+3A″) involved 17 valence shell electrons and the remaining 14 inner shell electrons were closed into 7 core orbitals (6A'+1A"). A total of 6471 ab initio energy points were generated from two grids defined in Jacobi coordinates and six additional grids defined around the important intermediates. For example, the $R_{A\text{-}BC}$, $r_{BC}$ and $\gamma_{A\text{-}BC}$ for the A-BC channel of ABC molecular are defined as the distance of the A atom relative to the center-of-mass of BC, the bond distance of BC and the angle between $R_{A\text{-}BC}$ and $r_{BC}$, respectively. In the P-O$_2$ channel, the grids were defined by $2.0 \le r_{OO}/a_0 \le 5.3$, $0 \le R_{P\text{-}OO}/a_0 \le 15.0$, and $0 \le \gamma_{P\text{-}OO}/\deg \le 90$. In the O-PO channel, the ranges were defined by $2.4 \le r_{PO}/a_0 \le 4$, $2.0 \le R_{O\text{-}PO}/a_0 \le 15.0$, and $0 \le \gamma_{O\text{-}PO}/\deg \le 180$. The additional grids around the equilibrium geometry, local minimum and transition states were constructed to be dense enough according to the geometry optimization (OPTG) (Eckert et al., 1997) results obtained from MOLPRO 2015 (Werner et al., 2020; Eckert et al., 1997).

**3 The CHIPR potential energy surface**

According to the spin-spatial Wigner-Witmer correlation, the dissociation scheme of the ground state $PO_2(X\,^2A_1)$ can be described by

$$PO_2(X\,^2A_1/^2A') \rightarrow O(^3P) + PO(X\,^2\textstyle\prod) \qquad\qquad R5$$
$$\rightarrow P(^4S) + O_2(X\,^3\Sigma^-) \qquad\qquad R6$$
$$\rightarrow P(^4S) + O(^3P) + O(^3P) \qquad\qquad R7$$

The ground state $PO_2(X\,^2A_1)$ dissociates adiabatically into $O(^3P) + PO(X\,^2\Pi)$, $P(^4S) + O_2(X\,^3\Sigma^-)$ and $P(^4S) + O(^3P) + O(^3P)$. Assuming the energy of infinitely separated $P(^4S) + O(^3P) + O(^3P)$ atoms to be the zero point, the global adiabatic CHIPR PES of the ground state $PO_2(X\,^2A_1)$ has the following many-body expansion (MBE) (Murrell, 1984; Varandas and Murrell, 1977) form:

$$V(\mathbf{R}) = V_{O_2}^{(2)}(R_1) + V_{PO}^{(2)}(R_2) + V_{PO}^{(2)}(R_3) + V_{PO_2}^{(3)}(\mathbf{R}) \qquad\qquad (1)$$

where $V^{(2)}$ are the two-body fragments represented by the diatomic potential energy curves (PECs) of $O_2(X\,^3\Sigma^-)$ and $PO(X\,^2\Pi)$. $V^{(3)}$ is the three-body fragment. In the CHIPR method (Rocha and Varandas, 2021, 2020; Varandas, 2013), two-body fragments are given by

$$V^{(2)}(R) = \frac{Z_A Z_B}{R} \sum_{k=1}^{L} C_k y^k \qquad\qquad (2)$$

where $Z_A$ and $Z_B$ are the nuclear charges of A and B atoms, respectively. $C_k$ are expansion coefficients of a $L^{th}$-order polynomial and $y$ is the basis set consisted of the linear combination of $R$-dependent functions [see Eq. (4) shown below]. For the AB$_2$-

type species like $PO_2$, the CHIPR three-body fragment can be simplified to the following permutation symmetric form (Rocha and Varandas, 2021, 2020; Varandas, 2013):

$$V^{(3)}(\mathbf{R}) = \sum_{i,j,k=0}^{L} C_{i,j,k} \left[ y_1^i \left( y_2^j y_3^k + y_2^k y_3^j \right) \right] \tag{3}$$

where $C_{i,j,k}$ are expansion coefficients for a $L^{th}$-order polynomial and $y_p$ are basis sets of coordinates ($p = 1, 2, 3$) for the reference geometry, which are expressed in terms of the $M^{th}$-order distributed-origin contracted basis set (Rocha and Varandas, 2021, 2020):

$$y_p = \sum_{\alpha=1}^{M-1} c_\alpha \phi_{p,\alpha} + c_M \phi_{p,M} \tag{4}$$

where

$$\phi_{p,\alpha} = \operatorname{sech}\left\{ \gamma_{p,\alpha} \left[ R_p - \zeta \left( R_p^{\mathrm{ref}} \right)^{\alpha-1} \right] \right\} \tag{5}$$

and

$$\phi_{p,M} = \left[ \frac{\tanh\left( 0.2 R_p \right)}{R_p} \right]^6 \operatorname{sech}\left\{ \gamma_{p,M} \left[ R_p - \zeta \left( R_p^{\mathrm{ref}} \right)^{M-1} \right] \right\} \tag{6}$$

are primitive bases. $\gamma_{p,\alpha}$ are non-linear parameters and $R_p^{\mathrm{ref}}$ represents the origin. The fitting process was carried out in the CHIPR 4.0 program (Rocha and Varandas, 2021). More detailed descriptions for the CHIPR method can refer to the related manuals (Rocha and Varandas, 2021, 2020). In recent years, there has been a lot of global PESs constructed based on this method, such as $H_3$ (Varandas, 2013), $HO_2$ (Varandas, 2013), $C_3$ (Rocha and Varandas, 2019a), $C_3H$ (Rocha and Varandas, 2019b), $PH_2$ (Chen et al., 2022), $NH_2$ (Li et al., 2022), $Si_2C$ (Li et al., 2023) and $SiC_2$ (Rocha et al., 2022).

### 3.1 Two-body fragment

Based on Eq (1), the PECs of the ground states $O_2(X\,^3\Sigma^-)$ and $PO(X\,^2\Pi)$ were fitted to the CHIPR form of Eq. (2) to obtain the two-body fragments of the MBE potential of $PO_2(X\,^2A_1)$. *Ab initio* potential energies were calculated at the MRCI(Q) level of theory with the AV5Z basis set. For $O_2(X\,^3\Sigma^-)$, $D_{2h}$ symmetry was chosen with the consideration of 10 active molecular orbitals ($3A_g + 1B_{3u} + 1B_{2u} + 3B_{1u} + 1B_{2g} + 1B_{3g}$) and 2 closed orbitals ($1A_g + 1B_{1u}$), while 10 active molecular orbitals ($6A_1 + 2B_1 + 2B_2$) and 6 closed orbitals ($4A_1 + 1B_1 + 1B_2$) of $C_{2v}$ symmetry were applied for $PO(X\,^2\Pi)$. A total of 43 and 34 *ab initio* energy points were obtained for $O_2(X\,^3\Sigma^-)$ and $PO(X\,^2\Pi)$, respectively.

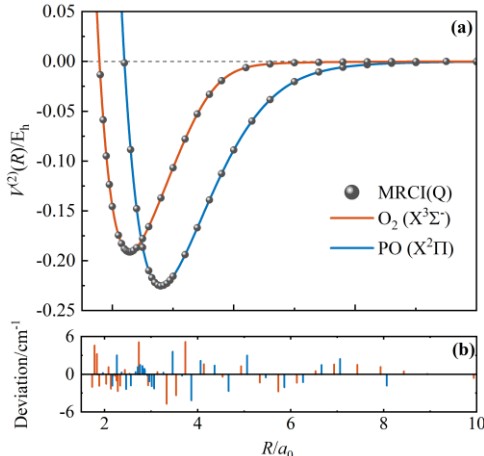

**Fig. 1 (a) The PECs of $O_2(X\,^3\Sigma^-)$ and $PO(X\,^2\Pi)$. The unit of potential energy is the atomic unit (Hartree, $E_h$). Solid lines are the CHIPR PECs. Solid circles are *ab initio* energies. (b) The deviations between *ab initio* energy points and the corresponding CHIPR energies.**

During the fitting, the $4^{th}$-order contracted bases [Eq. (4)] and $8^{th}$-order polynomial expression [Eq. (2)] were applied for $O_2(X\,^3\Sigma^-)$ and $PO(X\,^2\Pi)$. Fig. 1 (a) presents the final CHIPR PECs of $O_2(X\,^3\Sigma^-)$ and $PO(X\,^2\Pi)$ with the root-mean-square deviations (RMSDs) of 2.05 $cm^{-1}$ and 1.85 $cm^{-1}$, respectively. Fig. 1(b) shows the deviations between *ab initio* energy points and the corresponding CHIPR energies, which are within $\pm 6\ cm^{-1}$. The obtained CHIPR PECs of $O_2(X\,^3\Sigma^-)$ and $PO(X\,^2\Pi)$ reproduce well with the *ab initio* energy points and exhibit smooth strongly-bound valence region and reasonable dissociation behaviors. As shown in Table 1, the spectroscopic constants of our CHIPR PECs agree well with those of previous theoretical and

 experimental results. Hence, the CHIPR PECs of $O_2(X\,^3\Sigma^-)$ and $PO(X\,^2\Pi)$ are reliable and can be used as two-body fragments to construct a global PES of $PO_2(X\,^2A_1)$. The analytic functions of the $O_2(X\,^3\Sigma^-)$ and $PO(X\,^2\Pi)$ PECs are collected in the ready-to-use Fortran code in the Supplement.

**Table 1 Spectroscopic constants of the CHIPR PECs of $O_2(X\,^3\Sigma^-)$ and $PO(X\,^2\Pi)$, along with previous theoretical and experimental results.**

| Method | $R_e$ [a] | $D_e$ [b] | $\omega_e$ [c] | $\omega_e\chi_e$ [c] | $\alpha_e$ [c] | $B_e$ [c] |
|---|---|---|---|---|---|---|
| $O_2\,(X\,^3\Sigma^-)$ | | | | | | |
| CHIPR [d] | 2.285 | 5.203 | 1583.45 | 12.05 | 0.01545 | 1.4407 |
| Theory [e] | 2.281 | 5.220 | 1581.16 | 10.04 | 0.01254 | 1.4376 |
| Theory [f] | 2.282 | 5.100 | 1601 | | | |
| Exp. [g] | 2.282 | 5.213 | 1580.19 | 11.98 | 0.01593 | 1.4456 |
| | | | | | | |
| $PO\,(X\,^2\Pi)$ | | | | | | |
| CHIPR [d] | 2.802 | 6.130 | 1227.83 | 6.65 | 0.00540 | 0.7270 |
| Theory [e] | 2.787 | 6.221 | 1236.01 | 6.77 | 0.00574 | 0.7346 |
| Theory [h] | 2.801 | 6.076 | | | | |
| Exp. [g] | 2.789 | 6.15 | 1233.34 | 6.56 | 0.0055 | 0.7337 |

[a] The equilibrium geometry in unit of $a_0$. [b] The dissociation energy in unit of eV. [c] The units of $\omega_e$, $\omega_e\chi_e$, $\alpha_e$ and $B_e$ are cm$^{-1}$. [d] The CHIPR PECs. [e] Results at the icMRCI(Q)/CBS(56) + CV + DK level (Liu et al., 2014; Liu et al., 2017). [f] Results at the CCSDT/AVQZ level (Sordo, 2001). [g] Ref. (Huber and Herzberg, 1979). [h] Results at the MRCI(Q)-r/aug-cc-wCV5Z level (Prajapat et al., 2017).

 ## 3.2 Three-body fragment

**Table 2 The root-mean-square deviations (RMSDs) in the indicated energy range above the GM and those for the additional energy grids.**

| | $N$ [a] | Max deviation [b] | RMSD [c] | $N_{>RMSD}$ [d] |
|---|---|---|---|---|
| Energy Range [e] | | | | |
| 10 | 1560 | 95.4 | 24.6 | 447 |
| 20 | 1892 | 103.9 | 23.1 | 504 |
| 40 | 1986 | 152.0 | 24.4 | 485 |
| 60 | 2039 | 185.5 | 29.4 | 439 |
| 80 | 2131 | 226.5 | 37.9 | 339 |
| 100 | 2231 | 263.9 | 52.9 | 253 |
| 200 | 5911 | 280.3 | 84.4 | 1218 |
| 300 | 6415 | 319.1 | 90.1 | 1441 |
| 500 | 6471 | 387.9 | 91.5 | 1461 |
| Configuration [f] | | | | |
| GM | 1554 | 79.7 | 24.5 | 452 |
| LM | 490 | 71.4 | 21.4 | 140 |
| TS1 | 810 | 48.6 | 20.3 | 292 |
| TS2 | 405 | 29.8 | 10.5 | 118 |
| TS3 | 810 | 34.1 | 13.4 | 242 |
| TS4 | 810 | 60.8 | 14.5 | 188 |

[a] The number of energy points in the corresponding range. [b] The maximum deviation in the corresponding range, cm$^{-1}$. [c] The RMSD for the corresponding range, cm$^{-1}$. [d] The number of energy points with a deviation larger than the RMSD. [e] The indicated energy range above the GM, kcal mol$^{-1}$. [f] The additional energy grids.

For the CHIPR PES of $PO_2(X\,^2A_1)$, the 4th-order contracted bases [Eq. (4)] and 12th-order polynomial expression [Eq. (3)] were used to fit the three-body fragment. The first trial of the fitted PES presented the complex topographical features around  the reaction path, including the important intermediates of 1 global minimum (GM), 1 local minimum (LM) and 4 transition states (TS). Then, the OPTG was carried out for these configurations and the additional energy grids were calculated based on the OPTG results. During the subsequent fitting processes, the weights for the additional grids were set to be 5 and those for the remaining energy points were set to be 1. The final PES was constructed from 6471 *ab initio* energy points, covering an energy range up to 500 kcal/mol above the GM. Table 2 lists the RMSDs between the analytical CHIPR energies and *ab initio*  energies in the indicated energy range above the GM and those for the additional energy grids. The GM, LM and TSs of $PO_2(X\,^2A_1)$ were well reproduced by the CHIPR method with RMSDs lower than 25 cm$^{-1}$. The total RMSD is 91.5 cm$^{-1}$ (or 0.262 kcal/mol), implying the high fitting accuracy and reliability of the CHIPR PES of $PO_2(X\,^2A_1)$. The whole $PO_2(X\,^2A_1)$ PES is collected in the ready-to-use Fortran code in the Supplement.

## 4 Features of CHIPR PES

Fig. 2, Fig. 3 and Fig. 4 display the topographical features and the relevant stationary points of the CHIPR PES for the ground state $PO_2(X\ ^2A_1)$. Table 3 compares the main attributes of the stationary points for the CHIPR PES with other theoretical and experimental results, including the interatomic distances for OO ($R_1$) and PO ($R_2$ and $R_3$), the bond angle ($\theta$) between $R_2$ and $R_3$, the vibration frequencies (symmetrical stretching $\omega_1$, bending $\omega_2$ and antisymmetric stretching $\omega_3$) and the potential energies ($E$) relative to the $P(^4S) + O(^3P) + O(^3P)$ asymptote. In particular, our OPTG results for GM, LM, TS1, TS2, TS3 and

TS4 are also collected in Table 3, which are calculated at the MRCI(Q) level as implemented in MOLPRO 2015 (Werner et al., 2020). As shown in Table 3, the attributes of all the stationary points reproduced by CHIPR method are very similar to the OPTG results.

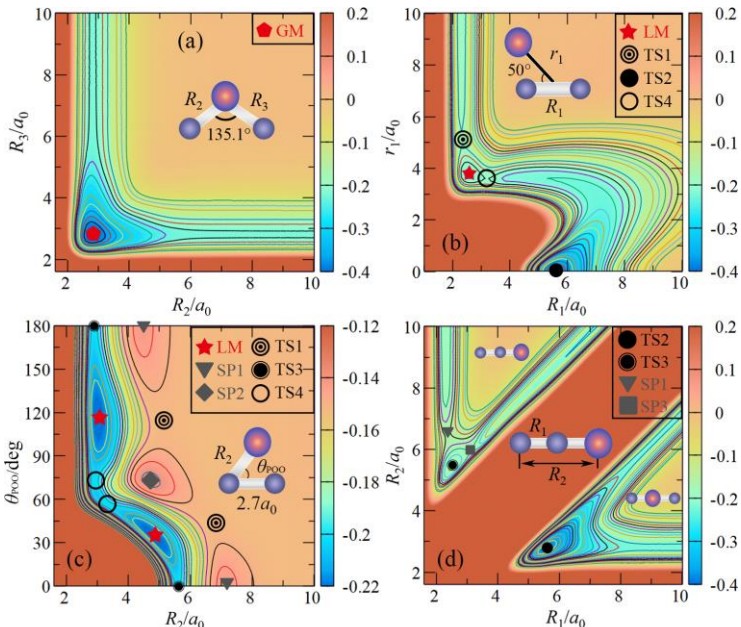

**Fig. 2 (a) The contour plot for bond stretching in the OPO bending configuration with $\theta$ fixed at 135.1°. Contours are equally spaced**
**by 0.02 $E_h$, starting from -0.4 $E_h$. (b) The contour plot for the insertion of the P atom into the $O_2$ fragment with the insertion angle of 50°. Contours are equally spaced by 0.011 $E_h$, starting from -0.38 $E_h$. (c) The contour plot for P moving around one of the O atoms with $R_1$ fixed at 2.7 $a_0$. Contours are equally spaced by 0.006 $E_h$, starting from -0.4 $E_h$. (d) The contour plot for bond stretching of OOP colinear configuration. Contours are equally spaced by 0.01 $E_h$, starting from -0.4 $E_h$.**

    Fig. 2 (a) shows the topographical features of the $PO_2(X\ ^2A_1)$ PES for the PO bond stretching from $\theta$ fixed at the equilibrium

angle of 135.1°. As shown in Fig. 2 (a), the GM of the CHIPR PES has a $C_{2v}$ symmetric configuration with bond lengths of $R_1$ = 5.142 $a_0$ and $R_2 = R_3$ = 2.788 $a_0$, which can be seen in Fig. 3 (b) and Fig. 4 as well. The vibrational frequencies are computed to be 1137.6 cm⁻¹, 464.7 cm⁻¹ and 1460.0 cm⁻¹ for $\omega_1$, $\omega_2$ and $\omega_3$, respectively. According to Table 3, these attributes of GM agree well with previous theoretical and experimental results, excepted that $\omega_3$ is slightly higher than experimental results and those calculated by the CCSD(T) method. Fig. 2 (a) presents the channel of an O atom dissociated from $PO_2(X\ ^2A_1)$ and the

corresponding dissociation energy $D_e$(O-PO) of the CHIPR PES is 5.305 eV, which is similar to the experiment value of 5.11 eV (Drowart et al., 1972) and the theoretical result of 5.24 eV at the CCSD(T)/(CBS+SO+DK+CV+ZPE) level (Bauschlicher, 1999). The dissociation energy of the P atom dissociated from $PO_2(X\ ^2A_1)$ is predicted to be 6.23 eV, showing a good concordance of the predicted value of 6.15 eV at CI/3-21G* level (Kabbadj and Liévin, 1989).

    Fig. 2 (b) and (c) show the entrance channel of $P(^4S) + O_2(X\ ^3\Sigma^-) \rightarrow O(^3P) + PO(X\ ^2\Pi)$, i.e. the contour plots for the insertion

of the P atom into the $O_2$ fragment with the insertion angle of 50° and for P moving around one of the O atoms with $R_1$ fixed at 2.7 $a_0$, respectively. As illustrated in these two figures, the MEP of the entrance channel is connected by TS1, LM and TS4 in turn, which is in accordance with the earlier prediction (Gomes et al., 2022; Douglas et al., 2019). The most important configuration for the reaction is the entrance barrier of TS1. The barrier height relative to the $P(^4S) + O_2(X\ ^3\Sigma^-)$ asymptote was theoretically predicted to be 0.032 eV at B3LYP/AVQZ level (Douglas et al., 2019) and 0.158, 0.142 and 0.137 eV at

MRCI(Q)/AVXdZ (X = T, Q, 5) levels (Gomes et al., 2022), respectively. Our CHIPR PES is fitted by the energies at MRCI(Q)/AV5Z level and the barrier height of TS1 is 0.133 eV, which is particularly consistent with the result at

MRCI(Q)/AV5dZ level (Gomes et al., 2022). Also, the attributes of TS1, LM, and TS4 showed good agreement with previous theoretical results, which are presented in details in Table 3.

**Table 3 Attributes of the global minimum (GM), local minimum (LM), transition state (TS) and second-order saddle (SP) of $PO_2(X\ ^2A_1)$ CHIPR PES**

| | Method | $R_1/a_0$ | $R_2/a_0$ | $R_3/a_0$ | $\theta$/deg | $\omega_1$/cm$^{-1}$ | $\omega_2$/cm$^{-1}$ | $\omega_3$/cm$^{-1}$ | $E$/eV |
|---|---|---|---|---|---|---|---|---|---|
| GM ($C_{2v}$) | CHIPR | 5.142 | 2.788 | 2.788 | 134.5 | 1137.6 | 464.7 | 1460.0 | -11.435 |
| | AVQdZ [a] | 5.118 | 2.767 | 2.767 | 135.3 | 1085.5 | 402.1 | 1495.7 | -11.437 |
| | Theory | 5.142 [b] | 2.785 [b] | 2.785 [b] | 134.8 [b] | 1073 [b] | 390 [b] | 1349 [b] | -11.40 [c] |
| | Theory [d] | 5.132 | 2.775 | 2.775 | 135.3 | 1081.2 | 391.6 | 1362.2 | |
| | Theory [e] | 5.123 | 2.786 | 2.786 | 133.7 | 1072.4 | 389.5 | 1316.8 | |
| | Theory [f] | 5.108 | 2.785 | 2.748 | 134.8 | 988.4 | 415.4 | 1316.4 | |
| | Exp. | 5.126 [g] | 2.771 [g] | 2.771 [g] | 135.3 [g] | 1117 [h] | 387 [h] | 1327.5 [i] | -11.26 [j] |
| | Exp. | | | | | 1075.4 [k] | 397.3 [k] | 1327.5 [l] | |
| LM ($C_s$) | CHIPR | 2.525 | 3.074 | 4.898 | 26.0 | 676.0 | 262.6 | 1009.9 | -6.135 |
| | AV5Z [a] | 2.522 | 3.075 | 4.857 | 26.7 | 709.4 | 228.9 | 1029.2 | -6.136 |
| | Theory [e] | 2.529 | 3.069 | 4.912 | 25.8 | 671 | 259 | 1016 | |
| | Theory [f] | 2.548 | 3.128 | 4.869 | 27.6 | 695.9 | 276.4 | 1010.5 | |
| TS1 ($C_s$) | CHIPR | 2.317 | 4.468 | 5.858 | 20.9 | 1412.4 | 269.4i | 277.1 | -5.070 |
| | AV5Z [a] | 2.317 | 4.361 | 5.827 | 20.5 | 1398.6 | 296.9i | 271.9 | -5.074 |
| | Theory [e] | 2.298 | 4.766 | 6.233 | 18.7 | 1497 | 121i | 225 | |
| | Theory [f] | 2.357 | 4.121 | 5.640 | 21.5 | 1253.2 | 426.5i | 306.9 | |
| TS2 ($D_{\infty h}$) | CHIPR | 5.548 | 2.774 | 2.774 | 180 | 975.9 | 434.9i | 1360.8 | -10.619 |
| | AVTZ [a] | 5.610 | 2.805 | 2.805 | 180 | 1030.4 | 588.3i | 1450.1 | -9.786 |
| | AV5Z [a] | 5.540 | 2.770 | 2.770 | 180 | | | | -10.621 |
| | Theory [m] | 5.420 | 2.710 | 2.710 [r] | 180 | 1199 | 994i | 1707 | |
| | Theory [n] | 5.446 | 2.723 | 2.723 [t] | 180 | | | | |
| TS3 ($C_{\infty v}$) | CHIPR | 2.475 | 2.917 | 5.392 | 0 | 1108.6 | 334.1i | 671.8 | -5.640 |
| | AV5Z [a] | 2.465 | 2.927 | 5.392 | 0 | | | | -5.592 |
| TS4 ($C_s$) | CHIPR | 2.922 | 2.900 | 4.187 | 44.2 | 1071.2 | 412.9 | 471.7i | -5.866 |
| | AV5Z [a] | 2.898 | 2.899 | 4.211 | 43.4 | 1051.3 | 398.6 | 556.0i | -5.866 |
| | Theory [e] | 2.976 | 2.881 | 4.038 | 47.4 | 1103 | 392 | 341i | |
| | Theory [f] | 3.130 | 2.918 | 4.797 | 39.1 | 956.2 | 351.9 | 640.6i | |
| SP1 ($C_{\infty v}$) | CHIPR | 2.356 | 4.027 | 6.384 | 0 | 401.9i | 286.6i | 1131.3 | -4.520 |
| SP2 ($C_{2v}$) | CHIPR | 2.397 | 4.225 | 4.225 | 33.0 | 618.7i | 415.2i | 1211.0 | -4.267 |
| SP3 ($C_{\infty v}$) | CHIPR | 3.009 | 5.849 | 2.840 | 0 | 638.6i | 219.4i | 1226.8 | -5.338 |

[a] Results from the geometry optimization (OPTG). [b] Results at the CCSD(T)/AVQZ level (Francisco, 2002). [c] Result at the CCSD(T)/(CBS+SO+DK+CV+ZPE) level (Bauschlicher, 1999). [d] Results at the CCSD(T)/AV5Z level (Liang et al., 2013). [e] Results at the B3LYP/AVQZ level (Douglas et al., 2019). [f] Results at the MRCI(Q)/AVTZ+d level (Gomes et al., 2022). [g] Equilibrium geometry deduced from the observed rotational constants (Kawaguchi et al., 1985). [h] The vibrational frequencies from laser induced fluorescence spectrum (Hamilton, 1987). [i] Origin of the $v_3$ fundamental band obtained from the infrared absorption spectrum (Qian et al., 1995). [j] The experimental atomization energy $D_0(PO_2)$ (Drowart et al., 1972). [k] Vibration frequencies from fluorescence emission and laser fluorescence excitation spectra (Lei et al., 2001). [l] Term value of the $v_3$ fundamental band obtained from laser absorption spectrum (Lawson et al., 2011). [m] Results at the SCF/3-21G* level (Kabbadj and Liévin, 1989). [n] Results at the SCF/6-31G* level (Lohr, 1984).

Fig. 2 (d) shows the contour plot for bond stretching of $R_1$ and $R_2$ at colinear configuration, including the OOP (upper left corner) and OPO (lower right corner) configurations. The notable features here are the OPO colinear transition state TS2 and POO colinear one TS3. The TS3 is predicted to locate at $R_1 = 2.475\ a_0$, $R_2 = 2.917\ a_0$ and $R_3 = 5.392\ a_0$, and is connected with two second-order saddle points (SP1 and SP3). The configuration of the colinear TS2 is $R_2 = R_3 = 2.774\ a_0$ and very close to GM. The potential energy of TS2 is 0.816 eV higher than that of GM, which is close to previous theoretical results of 1.05 eV and 0.93 eV at the CI/3-21G* level (Kabbadj and Liévin, 1989) and MP3/6-31G* level (Lohr, 1984), respectively. As shown in Table 3, the attributes of TS2 are similar to those of previous theoretical results (Kabbadj and Liévin, 1989; Lohr, 1984), except that the vibration frequencies are much less than those obtained at SCF/3-21G* level (Kabbadj and Liévin, 1989).

Fig. 3 (a) is the contour plot for the P atom moving around the center of mass of $O_2$ with $R_1$ fixed at 2.285 $a_0$, which displays the smooth long-range behavior of the CHIPR PES. There exist high barriers (SP1 and SP2) on the entrance channel of $P(^4S) + O_2(X\ ^3\Sigma^-)$ when the P atom inserts along the mid-perpendicular or molecular axis of $O_2$. When the Jacobi approaching angle is about $40 \sim 50°$, it is much easier for the P atom to cross the barrier (TS1) and reach to the LM, followed by the subsequent reaction process. At relative low collision energies, therefore, the approaching P atom will bond first with one of the two O atoms, rather than with them at the same time. Fig. 3 (b) shows the contour plot for O moving around the center of mass of PO with $R_2$ fixed at 2.806 $a_0$, exhibiting the exit channel of the $P(^4S) + O_2(X\ ^3\Sigma^-) \rightarrow O(^3P) + PO(X\ ^2\Pi)$ reaction. As shown in Fig.

3 (b), both TS3 and TS4 can be reached from the LM, but the dissociation of the system is relatively difficult to occur at TS3 due to the high barrier SP3 on the dissociation path. The two wells LM and GM represent the POO and OPO isomers, which are separated by the isomerization barrier TS4. The minimum exit channel is connected by the LM, TS4, GM and TS2 in turn, and then the system dissociates to the products of $O(^3P) + PO(X\ ^2\Pi)$.

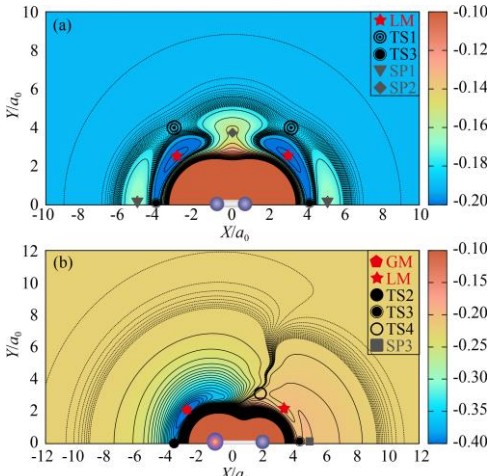

**Fig. 3 (a) The contour plot for P moving around the center of mass of $O_2$ with $R_1$ fixed at 2.285 $a_0$. Contours are equally spaced by 0.011 $E_h$, starting from -0.22 $E_h$. Dashed areas are contours equally spaced by 0.0005 $E_h$, starting from -0.192 $E_h$. (b) The contour plot for O moving around the center of mass of PO with $R_2$ fixed at 2.788 $a_0$. Contours are equally spaced by 0.013 $E_h$, starting from -0.41 $E_h$. Dashed areas are contours equally spaced by 0.0001 $E_h$, starting from -0.226 $E_h$.**

Fig. 4 depicts the relaxed triangular contour plot (Varandas, 1987) for the CHIPR PES of $PO_2(X\ ^2A_1)$ using the scaled hyperspherical coordinates ($\beta^*=\beta/Q$ and $\gamma^*=\gamma/Q$), given by

$$\begin{pmatrix} Q \\ \beta \\ \gamma \end{pmatrix} = \begin{pmatrix} 1 & 1 & 1 \\ 0 & \sqrt{3} & -\sqrt{3} \\ 2 & -1 & -1 \end{pmatrix} \begin{pmatrix} R_1^2 \\ R_2^2 \\ R_3^2 \end{pmatrix} \tag{7}$$

**Fig. 4 The relaxed triangular contour plot for the ground-state $PO_2$ in hyperspherical coordinates (see the definition in the text). Contours are equally spaced by 0.008 $E_h$, starting from -0.41 $E_h$.**

The hidden coordinate $Q$ (i.e. the sum of squares of the three bond distances) is allowed to relax to give the lowest potential energy, while $\beta$ and $\gamma$ define the shape of the triangle formed by the three atoms [see the work of Varandas (1987) for details]. It gives a better view of the major topographical features of the CHIPR PES, including the configurations and symmetries of all stationary points mentioned above. The MEP is connected by TS1, LM, TS4, GM and TS2 in turn. At high collision energies, the P atom is able to cross the $C_{2v}$ barrier SP2 and reach the GM directly, which will be confirmed in the following dynamic calculations. In this condition, the P atom approaches along the mid-perpendicular of $O_2$ and the two PO bonds stretch symmetrically, accompanied by the progressively open OPO angle, as shown in Fig. 4. Thus, after the P atom crosses over the SP2, the system will reach the symmetric GM instead of the asymmetric TS1, LM and TS4. Then, the system evolves through the linear transition state TS2 and finally dissociates to the products of $O(^3P) + PO(X\ ^2\Pi)$. There is also a collinear abstraction reaction path through TS3, which is difficult to occur due to the mutation of the molecular structure and the existence of

collinear barriers (SP1 and SP3) on the entrance and exit channels. We should reiterate that the critical intermediates were carefully verified by the OPTG results, the relevant *ab initio* energy grids were constructed to be dense enough and the CHIPR method reproduced them well. Hence, our CHIPR PES of $PO_2(X\,^2A_1)$ has reliable reaction channels and can be used to perform relevant dynamic calculations.

## 5 Dynamic calculations

Based on our CHIPR PES of $PO_2(X\,^2A_1)$, the reaction probability, ICS and rate constants of the $P(^4S) + O_2(X\,^3\Sigma^-) \rightarrow O(^3P) + PO(X\,^2\Pi)$ reaction were calculated using the QCT and TDWP methods. For each QCT calculations, the sampled trajectories, initial distance of the reactants and integration time step of classical equations of motion were 100000, 15 Å and 0.2 fs, respectively. The state-specified QCT ICS of $P(^4S) + O_2(X\,^3\Sigma^-; v, j) \rightarrow O(^3P) + PO(X\,^2\Pi)$ was calculated by (Li et al., 2014)

$$\sigma_r\left(E_{tr}; v, j\right) = \pi b_{max}^2 \frac{N_r}{N} \tag{8}$$

where $b_{max}$ is the maximum impact parameter, $N$ and $N_r$ represent the total and reactive trajectories, respectively. The state-specified and thermal QCT rate constants $k(T)$ were obtained by (Li et al., 2014)

$$k(T) = g_e(T)\left(\frac{8k_B T}{\pi\mu_{P+O_2}}\right)^{1/2} \pi b_{max}^2 \frac{N_r}{N} \tag{9}$$

and

$$k(T) = g_e(T)\left(\frac{2}{k_B T}\right)^{3/2}\left(\frac{1}{\pi}\right)^{1/2} Q_{vj}^{-1}(T)\sum_{vj}(2j+1) \\ \times \exp\left(-\frac{E_{vj}}{k_B T}\right)\int_0^\infty E_{tr}\sigma^x \exp\left(-\frac{E_{tr}}{k_B T}\right)dE_{tr} \tag{10}$$

where $\mu_{P+O_2}$ is the reduced mass of the reactants, $k_B$ is the Boltzmann constant, $Q_{vj}(T)$ is the ro-vibrational partition function for all the ro-vibrational states of $O_2(X\,^3\Sigma^-)$, $E_{vj}$ is the energy of the $(v, j)$ state and $E_{tr}$ is the translation energy. The rate constant was computed adiabatically for the ground state $PO_2(X\,^2A_1)$ and the electronic degeneracy factor $g_e(T)$ assumed the following form (Graff and Wagner, 1990):

$$g_e(T) = g_{PO_2}(T)\left(q_P(T)q_{O_2}(T)\right)^{-1} \tag{11}$$

where $g_{PO_2}(T) = 2$ is the degeneracy of the ground state $PO_2(X\,^2A_1)$, $q_P(T) = 4$ is the electronic partition function accounting for the fine structure of $P(^4S)$ and $q_{O_2}(T)$ takes into account two spin-orbit states for $O_2(X\,^3\Sigma^-)$, given by

$$q_{O_2(^3\Sigma^-)}(T) = g(^3\Sigma_{0^+}^-) + g(^3\Sigma_1^-)\exp(-\Delta/T) \tag{12}$$

where $g(^3\Sigma_{0^+}^-) = 1$, $g(^3\Sigma_1^-) = 2$ due to the doubly degenerate of $\Omega = \pm 1$ and $\Delta = 2.88$ K (2 cm$^{-1}$) is the spin-orbit splitting between $^3\Sigma_{0^+}^-$ and $^3\Sigma_1^-$ (Liu et al., 2014; Barrow and Yee, 1974).

During the TDWP calculations, the split-operator scheme was used to solve the Schrödinger equation. The Hamiltonian for the reactants (P and $O_2$) in the $PO_2$ system can be represented using the Jacobi coordinates:

$$H = -\frac{\hbar^2}{2\mu_R}\frac{\partial^2}{\partial R^2} + \frac{(\hat{J}-\hat{j})^2}{2\mu_R R^2} + \frac{\hat{j}^2}{2\mu_r r^2} + V(R, r) + h(r) \tag{13}$$

where $R$ is the distance of the P atom relative to the center-of-mass of $O_2$, $r$ is the bond distance of $O_2$, $\mu_R$ is the reduced mass between P and $O_2$, $\mu_r$ is the reduced mass of $O_2$, $\hat{J}$ is the total angular momentum and $\hat{j}$ is the rotational angular momentum of $O_2$, $V(R, r)$ is the Jacobi form of the CHIPR PES for $PO_2$ and $h(r)$ is the diatomic reference Hamiltonian, given by

$$h(r) = -\frac{\hbar^2}{2\mu_r}\frac{\partial^2}{\partial r^2} + V(r) \tag{14}$$

where $V(r)$ is the PEC of $O_2$. For the TDWP calculations on the adiabatic PES of $PO_2(X\,^2A_1)$, the time-dependent wave function was expanded by the body-fixed (BF) translational-vibrational-rotational basis:

$$\Psi_{v_0 j_0 K_0}^{JM\varepsilon}(R, r, t) = \sum_{n,v,j,K} F_{nvjK,v_0 j_0 K_0}^{JM\varepsilon}(t)u_n^v(R)\varphi_v(r)Y_{jK}^{JM\varepsilon}(R, r) \tag{15}$$

where $M$ and $K$ are the projection of $J$ onto the z axis of the space-fixed and BF coordinates, respectively; $u_n^v(R)$, $\varphi_v(r)$ and $Y_{jK}^{JK\varepsilon}$ are the translational basis, reference vibration eigenfunction for $O_2$ and total angular momentum eigenfunction, respectively;

$\varepsilon$ is the parity of the system; $n$ is the label of the translational basis and $(v_0, j_0, K_0)$ denotes the initial ro-vibrational state of $O_2$. The dynamic information is extracted from the final wave packet after a long propagation time. The reaction probability $P_{v_0 j_0 K_0}^J$ (E), total reaction cross section $\sigma_{v_0 j_0}(E)$ and temperature-dependent rate constant $k_{v_0 j_0}(T)$ can be calculated by

$$P_{v_0 j_0 K_0}^J (E) = \frac{\hbar}{\mu_r} \mathrm{Im} \left[ \psi(E) \left| \delta(r - r_0) \frac{\partial}{\partial r} \right| \psi(E) \right] \tag{16}$$

$$\sigma_{v_0 j_0}(E) = \frac{\pi}{(2 j_0 + 1)k^2} \sum_{K_0} \sum_J (2J + 1) P_{v_0 j_0 K_0}^J (E) \tag{17}$$

$$k_{v_0 j_0}(T) = g_e(T) \sqrt{\frac{8 k_B T}{\pi \mu_R}} \left( k_B T \right)^{-2} \int_0^\infty E \sigma_{v_0 j_0}(E) \exp\left( -\frac{E}{k_B T} \right) dE \tag{18}$$

where $k$ is the wavenumber of the initial state with the fixed collision energy $E$, and $k_B$ is the Boltzmann constant. Table 4 displays the parameters used in the TDWP calculations. The *J*-shifting method was used in the calculation and the reaction probabilities of $J = 0$ to 270 were obtained.

**Table 4 Parameters used in the TDWP calculations. All parameters are given in atomic units, except for the numbers of quantities or indicated otherwise.**

| Parameter | Value |
|---|---|
| Scattering coordinate (R) range | 0.01-13 |
| Number of translational basis functions | 450 |
| Internal coordinate (r) range | 1.6-9.6 |
| Number of vibrational basis functions | 285 |
| Number of rotational basis functions | 150 |
| Absorption region length in R / r | 3.0 / 3.0 |
| Absorption strength in R / r | 0.03 / 0.03 |
| Center of initial wave packet on scattering coordinate | 9.5 |
| K-block | 2 |
| Propagation time | 50000 |
| Time step for propagation | 10 |

Fig. 5 presents the CHIPR MEP for $P(^4S) + O_2(X\ ^3\Sigma^-) \rightarrow O(^3P) + PO(X\ ^2\Pi)$ along with the potential energies of the corresponding intermediates relative to the $P(^4S) + O_2(X\ ^3\Sigma^-)$ asymptote. This reaction is an exothermic reaction (-0.927 eV) with an intrinsic entrance barrier TS1 (0.133 eV) and two isomers (LM, GM) separated by the isomerization barrier TS4. Along the MEP, the P atom approaches $O_2$ at the Jacobi angle of about 40 ~ 50° and bonds with one of the two O atoms to form an unstable transition state TS1, followed by a transitory POO isomer LM. The system then evolves through TS4, the OPO isomer GM and the linear transition state TS2, accompanied by the progressively open OPO angle (i.e. 44.2° for TS4, 134.5° for GM and 180° for TS2). Finally, the linear $PO_2$ dissociates to the products of $O(^3P) + PO(X\ ^2\Pi)$. The structural diagrams of these intermediaries are also shown in Fig. 5, and the corresponding geometric parameters are given in Table 3.

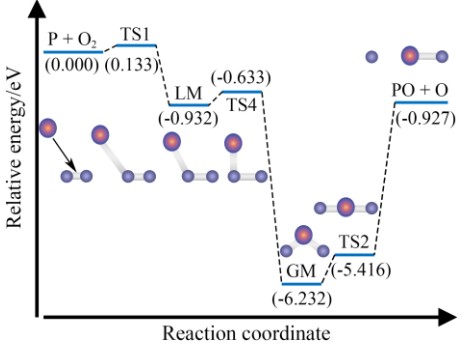

**Fig. 5 The minimum energy path (MEP) and relative intermediates of $P(^4S) + O_2(X\ ^3\Sigma^-) \rightarrow O(^3P) + PO(X\ ^2\Pi)$. Energies of intermediates relative to the $P(^4S) + O_2(X\ ^3\Sigma^-)$ asymptote are given in brackets and in unit of eV.**

The TDWP reaction probability of $P(^4S) + O_2(X\ ^3\Sigma^-; v=0, j=0) \rightarrow O(^3P) + PO(X\ ^2\Pi)$ at the total angular momentum $J = 0$, 80, 160 and 240 are presented in Fig. 6. For $J = 0$, the probability starts with a threshold and gradually rises to a peak. Subsequently, the probability decreases until the collision energy reaches 0.91eV, then it climbs again and eventually stabilizes. The threshold is about 0.1 eV, resulting from the $C_s$ barrier TS1 (0.133 eV) on the entrance channel. The secondary elevation after 0.91 eV is probably due to the opening of a new entrance channel, i.e. the P atom crosses the $C_{2v}$ barrier SP2 [0.936 eV relative to the $P(^4S) + O_2(X\ ^3\Sigma^-)$ asymptote] along the mid-perpendicular of $O_2$ and then reaches to GM directly, as discussed in the Section

4. Both the threshold and the starting point of the secondary elevation are slight smaller than the corresponding entrance barrier heights because of the quantum tunnelling effects. The trend of probability divides into three stages, as marked in Fig. 6. The first and third stages are the two ascending ones in the probability, showing the most common tendency of reactions controlled by barriers (TS1 and SP2). The second stage is the descending one after the peak, which is dominated by the exothermicity of the reaction.

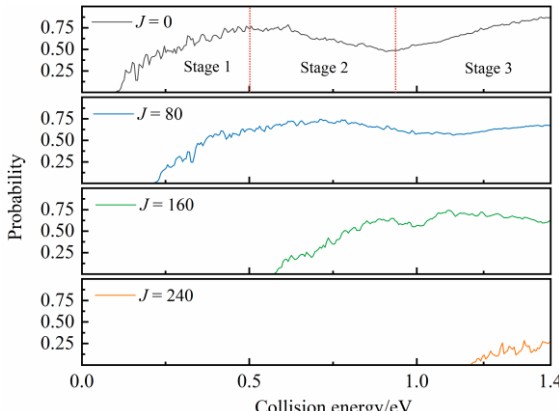

**Fig. 6 The TDWP reaction probabilities for P($^4$S) + O$_2$(X $^3\Sigma^-$; $v$=0, $j$=0) → O($^3$P) + PO(X $^2\Pi$) for $J$ = 0, 80, 160 and 240 as a function of the collision energy.**

For rotational cases ($J > 0$), the centrifugal effect appears and the rotational (or centrifugal) energy $J(J + 1)\hbar^2/(2\mu_R R^2)$ is added to the potential energy to obtain an effective potential (Waage and Rabinovitch, 1970). Since the centrifugal energy is positively correlated with $J$, the centrifugal barrier gradually increases with the increasement of $J$, resulting in the need for more translational energy to push the reactants over the barrier. Hence, the threshold shifts to larger collision energies with the increasement of $J$, as shown in Fig. 6.

The reaction probabilities obtained by the TDWP method are also oscillatory in all the three stages, which is the typical characteristic of quantum resonances. There are numerous bound and quasi-bound states which exist in the LM and GM potential wells, so the temporary reaction complexes are formed there under the bondage of potential well, leading to resonances. As shown in Fig. 5, the potential energy of the exit channel is 5.305 eV higher than GM leading to a deep potential well, although this reaction is exothermic. Hence, the bondage of potential well is strong at low collision energy, resulting in the long lifetimes of the collision complexes and the plenty of sharp and violent resonances before the peak (stage 1). Higher collision energy can help the complex get rid of the bondage of potential well and make the complex become short-living, which weakens the quantum resonance and makes the curves of probability smoother in stages 2 and 3.

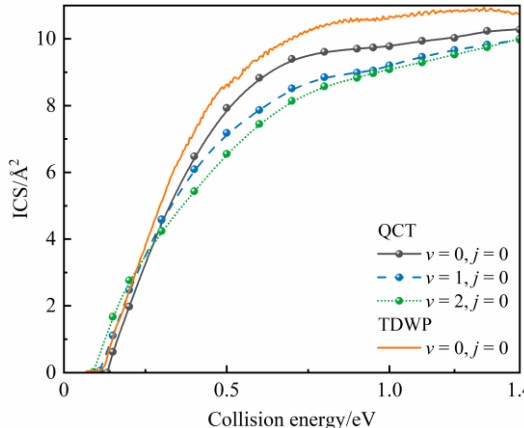

**Fig. 7 The state-specified QCT ($v$ = 0, 1 and 2) and TDWP ($v$ = 0) integral cross sections for the P($^4$S) + O$_2$(X $^3\Sigma^-$; $v$, $j$=0) → O($^3$P) + PO(X $^2\Pi$) reaction as a function of the collision energy.**

Fig. 7 displays the state-specified QCT ($v$ = 0, 1 and 2) and TDWP ($v$ = 0) ICSs for the P($^4$S) + O$_2$(X $^3\Sigma^-$; $v$, $j$=0) → O($^3$P) + PO(X $^2\Pi$) reaction as a function of the collision energy. The TDWP ICSs of vibrational excitation were not calculated due to the extremely expensive cost for this reaction involving three heavy atoms and deep potential wells. As shown in Fig. 7, both QCT and TDWP ICSs rise rapidly from the threshold in stage 1 and gradually reach a plateau (stage 2), and then increase

again and finally stabilize (stage 3). Also, the TDWP ICS at $v = 0$ is larger than the QCT ICS at $v = 0$ due to the quantum effects. The threshold is the minimum collision energy that the reaction can occur (i.e. the point of intersection of ICS and the X-axis), which are about 0.1 for TDWP ICS at $v = 0$ and 0.133, 0.11 and 0.09 eV for QCT ICSs at $v = 0$, $v = 1$ and $v = 2$, respectively. The threshold of QCT ICS at $v = 0$ is consistent with the entrance barrier height but the threshold of TDWP ICS at $v = 0$ is less than it, because the TDWP method includes the tunnelling effect and the zero-point-energy (ZPE) correction.

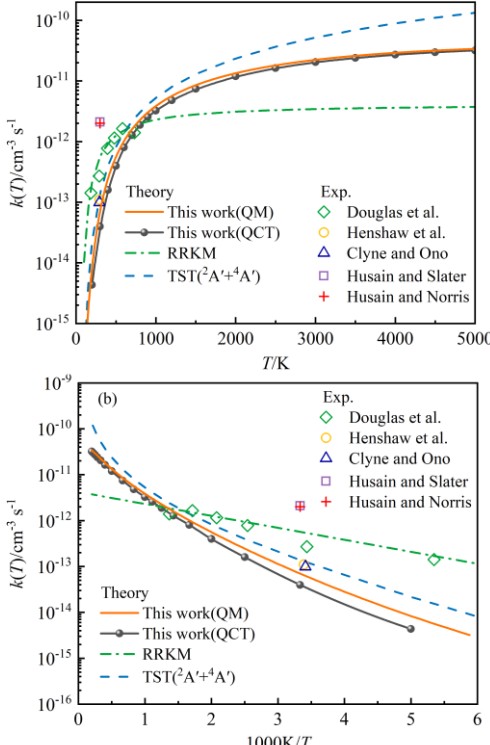

**Fig. 8 (a) The state-specified rate constant $k(T)$ for the $P(^4S) + O_2(X\ ^3\Sigma^-; v=0, j=0) \rightarrow O(^3P) + PO(X\ ^2\Pi)$ reaction as a function of temperature. (b) The corresponding Arrhenius plot. The green chain dotted line corresponds to the theoretical $k(T)$ (150 ≤ $T$/K ≤ 1400) of $P(^4S) + O_2$ based on the RRKM theory (Douglas et al., 2019). The blue dashed line corresponds to the theoretical rate constant of $P(^4S) + O_2(X\ ^3\Sigma^-) \rightarrow O(^3P) + PO(X\ ^2\Pi)$ based on the TST and carried out on MEPs of the $^2A'$ and $^4A'$ electronic states of $PO_2$ (Gomes et al., 2022). Diamond represents the experimental value using pulsed laser photolysis-laser-induced fluorescence technique (Douglas et al., 2019); Circle denotes the experimental value using discharge-flow method (Henshaw et al., 1987) ; triangle represents the experimental value using resonance-fluorescence detection in a discharge-flow system (Clyne and Ono, 1982); quadrate denotes the experimental value using resonance fluorescence method (Husain and Slater, 1978); cross represents the experimental value obtained from the attenuation of atomic resonance radiation in the vacuum ultraviolet (Husain and Norris, 1977).**

Furthermore, the vibrational excitation of $O_2$ has different effects on the reactivity for the three stages of this reaction. For the stage 1, the reactivity increases with increasing vibrational excitation, because the increased vibrational energy facilitates the reaction through the path with a barrier, resulting in less collision energy required for high vibrational states. Also, the threshold tends to decrease for the increasing vibrational excitations. For the stage 2, where the exothermicity dominates, the vibrational excitations of the reactants suppress the reaction reactivity. For the stage 3, both MEP and new entrance channel are contributing to the reaction, in which the reaction through MEP is dominated by the exothermicity like stage 2 and the reaction through the new entrance channel is dominated by the entrance barrier (SP2) like stage 1. In other words, the reaction through the new entrance channel is promoted by the vibrational excitation, so the slopes of QCT ICSs at $v = 1$ and $v = 2$ are significantly greater than QCT ICS at $v = 0$, as shown in Fig. 7. Moreover, the vibrational excitations of the reactants suppress the reactivity of the reaction through MEP and the reaction mainly occurs in this channel. Hence, the combined effect of the vibrational excitation for the two channels in stage 3 is that the vibrational excitations of the reactants suppress the reaction reactivity.

Fig. 8 presents the state-specified QCT and TDWP rate constants for $P(^4S) + O_2(X\ ^3\Sigma^-; v=0, j=0) \rightarrow O(^3P) + PO(X\ ^2\Pi)$ versus the temperature ranging below 5000 K. This reaction is exothermic and dominated by the entrance barrier TS1, so the rate constants show a significant Arrhenius linear behaviour at relatively low temperatures, as shown in Fig. 8 (b). At high temperatures, the rate constants deviate from the Arrhenius behaviour, because the effect of the barrier is weakened and the reaction activation energy becomes temperature-dependent. As expected, the TDWP rate constant is higher than QCT rate constant due to the quantum effects, especially at low temperatures.

Previous experimental rate constants are also given in Fig. 8 for comparison. Obviously, the rate constants at room temperature determined by various experiments differ by almost an order of magnitude. The earliest two experiments produced extremely large rate constants (Husain and Norris, 1977; Husain and Slater, 1978) and such large values was attributed to the large amounts of secondary dissociation products (Clyne and Ono, 1982). The subsequent experiments were carefully performed to

minimize this possible effect and obtained relatively lower rate constants (Clyne and Ono, 1982; Henshaw et al., 1987; Douglas et al., 2019), in which the recent experiment presented the temperature-dependent rate constants (Douglas et al., 2019). As shown in the Arrhenius plot of Fig. 8 (b), however, the recent experimental rate constants do not exhibit a good linear behaviour, unless the values at maximum and minimum temperatures are excluded. Therefore, four experimental rate constants at about 300-600 K are expected to be more reliable. As shown in Fig. 8, the present QCT and TDWP rate constants are lower than the

experimental values. It is partly because the experimental results may be suffering from secondary chemistry, i.e. some O atoms and PO molecules could be produced from reactions of $P(^2P, ^2D) + O_2$, albeit at extremely low quantities.

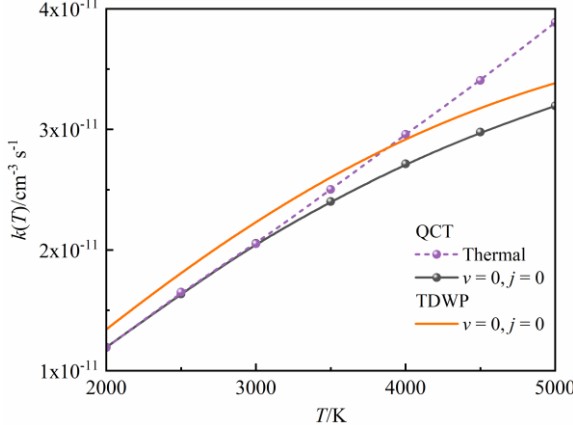

**Fig. 9 The state-specified ($v = 0, j = 0$) QCT and TDWP rate constants and thermal QCT rate constant for the $P(^4S) + O_2(X\,^3\Sigma^-) \rightarrow O(^3P) + PO(X\,^2\Pi)$ reaction as a function of temperature.**

The theoretical rate constants based on the RRKM theory (Douglas et al., 2019) and TST (Gomes et al., 2022) are also shown in Fig. 8. The RRKM rate constant available for temperatures of 150-1400 K was optimized by the experimental data and the TST rate constant was fitted from the calculated values at the temperatures below 1000 K. The rate constant obtained by TST includes the contribution of the doublet and quartet electronic states of $PO_2$ corresponding to $P(^4S) + O_2(X^3\Sigma^-)$, in which the quartet state works mainly at temperatures above about 600 K (Gomes et al., 2022). We did not consider the contribution from

the quartet state due to the extremely high computational cost, therefore our rate constants at very high temperatures may be slightly underestimated. However, the entrance barrier height including ZPE correction for the quartet state (0.3 eV) is about three times above that of the doublet state (0.105 eV) (Gomes et al., 2022), so the effect of quartet state on the rate constant may be small. At low temperatures, the RRKM rate constant is much higher than the TST rate constant and our QCT and TDWP rate constant, which can be explained by the fact that the low-temperature reactivity of such barrier-dominated reaction

like $P(^4S) + O_2(X\,^3\Sigma^-) \rightarrow O(^3P) + PO(X\,^2\Pi)$ depends sensitively on the entrance barrier height. The entrance barrier height adopted when calculating the RRKM rate constant was obtained to be 0.032 eV at B3LYP/AVQZ level (Douglas et al., 2019). This barrier height was also predicted to be 0.158, 0.142 and 0.137 eV at the MRCI(Q)/AVXdZ (X = T, Q, 5) levels (Gomes et al., 2022), in which the MRCI(Q)/AVTdZ result was modified to be 0.105 eV by including ZPE correction, and then used in the TST study (Gomes et al., 2022). There is evidence that the B3LYP method is not good at predicting the barrier heights

on the reaction path (Zhao and Truhlar, 2008; Peverati and Truhlar, 2012). Our CHIPR PES is fitted by the potential energies at MRCI(Q)/AV5Z level and the entrance barrier height is 0.133 eV, which is particularly consistent with the result at MRCI(Q)/AV5dZ level (Gomes et al., 2022), so the CHIPR PES is accurate and reliable. The relatively small QCT rate constants at low temperatures are due to the ZPE problem in the QCT treatment. The TDWP rate constants consider the ZPE correction and the tunnelling through the entrance channel barrier and the threshold of ICS is about 0.1 eV, agreeing with the

modified barrier height (0.105 eV) used in the TST calculation. The difference of the barrier height (0.05 eV) may be due to the tunnelling effect.

At high temperatures, it is necessary to point out that the previous two theoretical Arrhenius formulas (Douglas et al., 2019; Gomes et al., 2022) were fitted without high-temperature rate constants backing. Since the rate constant will deviate from the Arrhenius behaviour at high temperatures, the RRKM and TST rate constants remains to be verified at high temperatures. Moreover, the rate constants obtained by RRKM theory are obviously lower than our QCT and TDWP rate constants at high temperatures, because the original Arrhenius formula [$4.2 \times 10^{-12} \exp(-600/T)$] applied for the RRKM rate constants assumes that the reaction activation energy is temperature-independent, which is not applicable at high temperatures. Note that the population of the ro-vibrational excited $O_2$ increases at high temperatures, which could affect the reaction activity according to the analysis of ICS above. Fig. 9 compares the state-specified ($v = 0, j = 0$) QCT and TDWP rate constants and thermal QCT rate constant for $P(^4S) + O_2(X\ ^3\Sigma^-) \rightarrow O(^3P) + PO(X\ ^2\Pi)$ versus the temperature ranging from 2000 to 5000 K. The thermal TDWP rate constant was not calculated due to the extremely expensive cost. As shown in Fig. 9, the ro-vibratioal ground state ($v = 0, j = 0$) $O_2$ plays an absolute dominant role at temperatures below 3000 K, while the ro-vibrational excited $O_2$ appears at temperatures above 3000 K and promotes the reactive activity. Hence, our TDWP rate constant is reliable at temperatures below 3000 K, while it probably underestimates the rate constant at temperatures above 3000 K due to the neglect of the ro-vibrational excitation of $O_2$.

The rate constants of TDWP ($v = 0, j = 0$), QCT ($v = 0, j = 0$) and QCT (Thermal) for the $P(^4S) + O_2(X^3\Sigma^-) \rightarrow O(^3P) + PO(X^2\Pi)$ reaction can be approximated using the three-parameter Kooij function, given by (Laidler, 1996)

$$k(T) = A\left(\frac{T}{300}\right)^{\alpha} e^{-\beta/T} \tag{19}$$

where $A$, $\alpha$, and $\beta$ are fitting parameters. The rate constant curves are divided into four temperature ranges, and the fitting parameters are summarized in Table 5. The fitted rate constants deviate less than 1 per cent from our calculated ones. These rate constants are calculated in the atom-diatom system and independent of pressure, which may be applicable to the ISM and upper planetary atmospheres.

**Table 5 Parameters for Kooij function obtained by fitting the computed rate constants.**

| Method | $T$ / K | $A$ / $cm^3s^{-1}$ | $\alpha$ | $\beta$ / K |
|---|---|---|---|---|
| TDWP ($v = 0, j = 0$) | 100 - 600 | $2.2508 \times 10^{-12}$ | 1.3554 | 1041.9940 |
| | 600 - 1000 | $4.4682 \times 10^{-12}$ | 0.9430 | 1283.2565 |
| | 1000 - 2000 | $8.6398 \times 10^{-12}$ | 0.6532 | 1596.5685 |
| | 2000 - 5000 | $6.7105 \times 10^{-11}$ | -0.0146 | 3193.4739 |
| QCT (Thermal) | 200 - 600 | $4.7988 \times 10^{-13}$ | 2.5132 | 734.5248 |
| | 600 - 1000 | $6.7196 \times 10^{-12}$ | 0.7068 | 1574.9928 |
| | 1000 - 2000 | $6.7933 \times 10^{-12}$ | 0.7215 | 1610.7254 |
| | 2000 - 5000 | $2.0656 \times 10^{-12}$ | 1.0844 | 590.6460 |
| QCT ($v = 0, j = 0$) | 2000 - 5000 | $5.7104 \times 10^{-11}$ | 0.0244 | 3237.2285 |

## 6 Atmospheric Implications

The ablation of IDPs in the upper atmosphere (mainly at heights between 70 and 110 km) of terrestrial planets delivers about 6,200 kg yr$^{-1}$ ablated phosphorus to the atmosphere (Carrillo-Sánchez et al., 2020), where the temperatures ranging from below 100 to over 2500 K (in ablation IDPs). Several reaction networks of meteor-ablated phosphorus in the Earth's upper atmosphere (Douglas et al., 2019; Douglas et al., 2020; Plane et al., 2021) indicate that the initial oxidation of P will proceed through the successive oxidation by $O_2$ to produce OPO (i.e. reactions R1 and R2). The oxidation by $O_3$ is not significant because $O_2$ is $10^5$ times more abundant than $O_3$ at this altitude. Also, OPO is likely dissociated into PO and P as a result of hyperthermal collisions with air molecules (Carrillo-Sánchez et al., 2020). Therefore, the $P + O_2 \rightarrow O + PO$ reaction may occur throughout the upper mesosphere and thermosphere. Our rate constants are fitted by sufficient data below 5000 K, which is appropriate for most altitude of the Earth's atmosphere and can be used to model its phosphorus chemistry.

A recent study developed a reaction network of phosphorus atmosphere chemistry (Plane et al., 2021), including the possible routes from $P + O_2 \rightarrow O + PO$ to the stable reservoirs ($H_3PO_3$ and $H_3PO_4$). Subsequently, they incorporated the rate constants of the associated reactions into the Whole Atmosphere Community Climate Model from the US National Center for

Atmospheric Research (Gettelman et al., 2019), and then explored the vertical profiles of the P-containing species and the global mean P deposition flux. Also, they estimated that the fraction of the ablated phosphorus forming bioavailable metal phosphites was 11%.

One of our concerns is that the theoretical predicted rate constants of $P + O_2 \rightarrow O + PO$ diverge from experimental result at 200 K, which is about the typical temperature of the upper mesosphere and lower thermosphere. For the reaction with entrance barrier, its rate constant usually has Arrhenius linear behaviour at relative low temperatures. Our QCT and TDWP rate constants, as well as the TST rate constant (Gomes et al., 2022), exhibit highly consistent Arrhenius linear behaviour at relative low temperatures, with slopes close to those of the experimental rate constants at about 300-600 K (Douglas et al., 2019), as

shown in Fig. 8 (b). There is reason to believe that the experimental results of 300-600K are very reliable., Whereaswhereas, the experimental rate constant at about 200 K diverges from this linear behaviour and seems to be slightly overestimated., so Therefore, the lifetime of the P atoms in the atmosphere may be a little longer than previously thought (Carrillo-Sánchez et al., 2020). Further experiments for the rate constants at approximately 200 K of $P + O_2 \rightarrow O + PO$ are encouraged in the future.

## 7 Conclusions

In this work, we have constructed a global CHIPR PES of the ground state $PO_2(X\ ^2A_1)$ based on a total of 6471 *ab initio* energy points computed at the MRCI(Q)/AV5Z level. The *ab initio* grids for the critical intermediates are constructed to be dense enough based on the OPTG results. The total RMSD of the CHIPR PES is 91.5 cm$^{-1}$ and the RMSDs for the critical intermediates are lower than 25 cm$^{-1}$. The PES presents complex topographical features with multiple potential wells and barriers. The long-range interactions, diatomic potentials and dissociation energies of each asymptotic channel are reasonably

reproduced. The attributes of the main intermediates agree well with available experimental and theoretical results as well as our OPTG results. The corresponding adiabatic MEP of $P(^4S) + O_2(X\ ^3\Sigma^-) \rightarrow O(^3P) + PO(X\ ^2\Pi)$ features the barrier insertion of the P atom into the $O_2$ bond at the Jacobi angle of about $40 \sim 50°$, and the reaction then evolves through LM, TS4, GM and TS2 in turn. Based on the CHIPR PES, the state-specified reaction probability and ICSs for this reaction are calculated using QCT and TDWP methods. The results show three stages of reaction controlled by barrier TS1 (stage 1), exothermicity (stage

2) and barrier SP1 (stage 3) in turn. The vibrational excitation of $O_2$ promotes the reaction in stages 1 and 3, but suppresses the reactivity in stage 2. Meanwhile, the state-specified and thermal rate constants are predicted for the temperatures ranging from 200 to 5000 K, and then compared with available experimental and theoretical results. The rate constants show a significant Arrhenius linear behavior at relatively low temperatures, but they deviate from the Arrhenius behavior at high temperatures. The ro-vibratioal ground state $O_2$ plays an absolute dominant role at temperatures below 3000 K, while the ro-vibrational

excitation of $O_2$ promotes the reactive activity at temperatures above 3000 K.

The presented CHIPR PES of $PO_2(X\ ^2A_1)$ can be used for molecular simulations of reactive or non-reactive collisions and photodissociation of the $PO_2$ system in atmospheres and interstellar medium. Moreover, this analytical PES of $PO_2$ can provide reliable two-body fragments and three-body fragment for the construction of $PO_3$ and $HPO_2$ PESs using MBE form, so as to carry out dynamic study of reactions R2 and R3, which are also important for generating $H_3PO_3$ in the Earth's atmosphere. The

485 computed reaction probability, ICSs and rate constants may help to explain the relevant thermochemical reactions in related atmospheric and interstellar media.

**Data availability.** Additional relevant data and supporting information are given in the Supplement. All data used in this study are available upon request from the corresponding authors.

**Supplement.** The ready-to-use Fortran code for the whole CHIPR PES of $PO_2(X\ ^2A_1)$ is given in the Supplement, including

490 the CHIPR function and all the coefficients of the $O_2(X\ ^3\Sigma^-)$ and $PO(X\ ^2\Pi)$ PECs and three-body fragment.

**Competing interests.** We declare that we have no conflict of interest.

**Acknowledgements.** This work is supported by the National Natural Science Foundation of China (Grant no. 52106098) and

the Natural Science Foundation of Shandong Province (Grant no. ZR2021QE021). Zhi Qin also acknowledges the Postdoctoral Innovation Project of Shandong Province and the Postdoctoral Applied Research Project of Qingdao City. The scientific calculations in this paper have been done on the HPC Cloud Platform of Shandong University.

**Author contribution.** ZQ and GC designed the concept of the study, carried out the calculations, interpreted the results, and prepared the paper. XL helped with data analysis. LL supervised the study and acquired the primary funding to support this research. All of the authors discussed the scientific findings and provided valuable feedback for manuscript editing.

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
