# Peer review of "Reaction dynamics of $P({}^{4}S) + O_{2}(X {}^{3}\Sigma^{-}) \rightarrow O({}^{3}P) + PO(X {}^{2}\Pi)$ on a global CHIPR potential energy surface of $PO_{2}(X {}^{2}A_{1})$ : implication for atmospheric modelling"

_EGUsphere, 2023_

## Author Comment (AC1)

**Response to the reviewer 1**

The manuscript presents a computational investigation into the reaction dynamics of the $P(^4S) + O_2 \rightarrow$ PO + O system. Several thousand high level ab initio energy calculations, carried out using the multireference configuration interaction [MRCI(Q)] method with an aug-cc-pV5Z basis set, were carried out to map out the PES. These energy points were then fit using the combined-hyperbolic-inverse-power -representation (CHIPR) method in order to construct a global PES, from which features of the PES such as global and local minima and transitions states (TSs), as well as minimum energy pathway for the reaction, were identified. Using the CHIPR PES, rate coefficients for the reaction were calculated using the quasi-classical trajectory (QCT) and time-dependant wave-packet (TDWP) methods. Rate coefficients for the reaction of vibrationally excited $O_2$ with $P(^4S)$ were also calculated.

The reaction $P(^4S) + O_2 \rightarrow$ PO + O is important to both astrochemistry, where it is likely an important source of PO in the interstellar medium, and to atmospheric chemistry, being the first step in the oxidation of P atoms from the ablation of interstellar dust particles. As such, accurate rate coefficients over a wide range of temperatures are required to accurately model the P chemistry in these environments. There have been two previous computational studies of the reaction, by Douglas et. al. (2019) who calculate the surface at the B3LYP level and use transition state theory (TST) to fit the surface to experimental data, and by Gomes et. al. (2022), who calculate the surface at the MRCI(Q) level of theory, and also use TST to predict temperature dependant rate coefficients (I believe up to around 1000 K). The justification for the current work is twofold; firstly that the B3LYP method employed by Douglas et. al. (2019) is known to underestimate barrier heights, and secondly the TST theory employed by both studies may not provide accurate rate coefficients due to unincluded non-statistical effects, making the case for a dynamical study to be carried out.

The work presented in the manuscript is of a high quality, and the results and conclusions are generally well explained. I have several minor comments which I list below, as well several technical corrections I have spotted.

**Reply:** We thank the reviewer for their comments which we address below.

**Specific Comments:**

**#Comment 1:** Line 256 onwards and Figure 4 – I don't fully understand what the x and y axis on Figure 4 physically represent. Some more detail on how to interpret the relaxed triangular contour plot and what physical dimensions the x and y axis relate to would be good.

**Reply:** Thanks for your advice.

An important initial step in interpreting an elementary chemical reaction, as well as the associated energy transfer and disposal, is to do a graphical analysis of the relevant PES and to identify the stationary points. Unfortunately, obtaining a complete picture of the triatomic PES is difficult due to the fact that the PES is mathematically a hypersurface in four-dimensions (i.e. three spatial coordinates $R_i$ and one energy coordinate). Most contour plots are drawn by fixing one degree of freedom, so as to simplify the PES into three-dimensional ones, such as Figs. 2 and 3. These contour plots only present local topographical features and partial stationary points of the PES. The triangular plot of the triatomic PES preserves the problem's full permutational symmetry, in which the permutation symmetry of a triatomic molecule is best visualized graphically using symmetry coordinates that convert as irreducible representations of the molecule's symmetry point group, as shown in Eq. (7):

$$\begin{pmatrix} Q \\ \beta \\ \gamma \end{pmatrix} = \begin{pmatrix} 1 & 1 & 1 \\ 0 & \sqrt{3} & -\sqrt{3} \\ 2 & -1 & -1 \end{pmatrix} \begin{pmatrix} R_1^2 \\ R_2^2 \\ R_3^2 \end{pmatrix}$$ (1)

Using the $D_{3h}$ classification, $Q$ is associated with the totally symmetric representation $A_1'$, and the pair $(\beta, \gamma)$ is associated with the doubly degenerate representation E'. The hidden coordinate $Q$ (i.e. the sum of squares of the three bond distances) is allowed to relax to give the lowest potential energy and $\beta$ and $\gamma$ define the shape of the triangle formed by the three atoms, as shown in Fig. R1. Fig. 4 is drawn using a normalized coordinate system $(1, \beta^*=\beta/Q, \gamma^*=\gamma/Q)$, so the contour is plotted in a circle of radius 1 and the center of the circle represents an equilateral triangle shape. For more detailed definitions, see this paper [Varandas A J C. A useful triangular plot of triatomic potential energy surfaces. Chemical physics letters, 1987, 138(5): 455-461.].

The structures of the stationary points are shown in details in the local contour plots, as shown in Figs. 2 and 3. However, the reaction path in these local contour plots is incomplete, only including entrance or exit

channels. The main purpose of the triangular plot is presenting all the stationary points, so as to clearly show the whole reaction paths on the PES, although the structures of the stationary points might not be intuitive.

[Figure]

**Fig. R1 Symmetry coordinates and their relation to molecular conformation. The italics show the locus of the possible symmetry point groups assuming the nuclei to be identical. Where unassigned, the symmetry point group will be Cs.**

This triangular plot is widely used in the construction of the global PESs of triatomic molecules. However, it takes a lot of space to clearly explain it. Explaining the triangular plot is not the gist of this article. Hence, we have added some brief statements, as shown below:

Lines 243 to 244 - 'The hidden coordinate $Q$ (i.e. the sum of squares of the three bond distances) is allowed to relax to give the lowest potential energy, while $\beta$ and $\gamma$ define the shape of the triangle formed by the three atoms [see the work of Varandas (1987) for details].'

**#Comment 2:** Line 322 – 'The system then evolves through TS4, the OPO isomer GM and the linear transition state TS2, accompanied by the progressively open $\theta$.' Do you mean to say that as you progress from the GM to TS2 the OPO angle increases?    I think this needs to be made clearer.

**Reply:** Thanks for your advice. We have rephrased these sentences, as shown below:

Lines 308 to 311 - 'The system then evolves through TS4, the OPO isomer GM and the linear transition state TS2, accompanied by the progressively open OPO angle (i.e. 44.2° for TS4, 134.5° for GM and 180° for TS2). Finally, the linear $PO_2$ dissociates to the products of $O(^3P) + PO(X\ ^2\Pi)$. The structural diagrams of these intermediaries are also shown in Fig. 5, and the corresponding geometric parameters are given in Table 3.'

**#Comment 3:** Line 331 – you describe the secondary elevation after 0.91 eV (stage 3) as probably due to the opening of a new entrance channel, with the P atom crossing over the second-order saddle point SP2 to reach the GM directly. I'm guessing this refers to the contour plot in Figure 2c, which I think you should refer to here. However, it appears that when the P atom approaches the $O_2$ at an angle of 90° (along the mid-perpendicular?) it reaches the LM rather than the GM. Is this the case? I think this requires slightly more explaining.

**Reply:** Thanks for this suggestion. When the P atom approaches $O_2$ at an angle of 90°, the two PO bonds will stretch symmetrically (i.e. the geometry of the system is like an isosceles triangle) and the OPO angle will expand. Thus, after the P atom crossing over the second-order saddle point SP2, the system will reach the symmetric GM instead of the asymmetric LM. Figure 2c cannot show this entrance channel, because it fixes the OO bond length and the two PO bonds are asymmetrical. We have added more explaining for this entrance channel, as shown below:

Lines 246 to 251 - 'At high collision energies, the P atom is able to cross the $C_{2v}$ barrier SP2 and reach the GM directly, which will be confirmed in the following dynamic calculations. In this condition, the P atom approaches along the mid-perpendicular of $O_2$ and the two PO bonds stretch symmetrically, accompanied by the progressively open OPO angle, as shown in Fig. 2. Thus, after the P atom crosses over the SP2, the system will reach the symmetric GM instead of the asymmetric TS1, LM and TS4. Then, the system evolves through the linear transition state TS2 and finally dissociates to the products of $O(^3P) + PO(X\ ^2\Pi)$.'

Lines 318 to 321 - 'The secondary elevation after 0.91 eV is probably due to the opening of a new entrance channel, i.e. the P atom crosses the $C_{2v}$ barrier SP2 [0.936 eV relative to the $P(^4S) + O_2(X\ ^3\Sigma^-)$ asymptote] along the mid-perpendicular of $O_2$ and then reaches to GM directly, as discussed in the Section 4.'

**#Comment 4:** Line 338 – the last sentence of this paragraph I think need more explanation (i.e. how rotational excitation creates a barrier / increases the threshold energy).

**Reply:** This is indeed an important point worth mentioning. We have added more explanations for the centrifugal effect, as shown below:

Lines 329 to 333 - 'For rotational cases ($J > 0$), the centrifugal effect appears and the rotational (or centrifugal)

energy $J(J + 1)\hbar^2/(2\mu_R R^2)$ is added to the potential energy to obtain an effective potential (Waage and Rabinovitch, 1970). Since the centrifugal energy is positively correlated with $J$, the centrifugal barrier gradually increases with the increasement of $J$, resulting in the need for more translational energy to push the reactants over the barrier. Hence, the threshold shifts to larger collision energies with the increasement of $J$, as shown in Fig.6.'

**#Comment 5:** Around line 343 – the resonances you are referring to are in stage 1 in Figure 6? I think state this for clarity.

**Reply:** Thanks for this suggestion. The resonances are in all the three stages and the resonance strength is related to the life time of the reaction intermediate (collision complex). We have added more explanations for the resonance, as shown below:

Lines 334 to 341 - 'The reaction probabilities obtained by the TDWP method are also oscillatory in all the three stages, which is the typical characteristic of quantum resonances. There are numerous bound and quasi-bound states which exist in the LM and GM potential wells, so the temporary reaction complexes are formed there under the bondage of potential well, leading to resonances. As shown in Fig. 5, the potential energy of the exit channel is 5.305 eV higher than GM leading to a deep potential well, although this reaction is exothermic. Hence, the bondage of potential well is strong at low collision energy, resulting in the long lifetimes of the collision complexes and the plenty of sharp and violent resonances before the peak (stage 1). Higher collision energy can help the complex get rid of the bondage of potential well and make the complex become short-living, which weakens the quantum resonance and makes the curves of probability smoother in stages 2 and 3.'

**#Comment 6:** Around line 354 onwards – This sentence is a little confusing as you're initially referring to the absolute ICS values (for which the TDWP values are greater), and then later the threshold values (for which the QCT value is greater). Maybe split into two. When you state that the QCT ICS values are less than the TDWP ICS values, I think you need to state that this is for $v = 0$. And when talking about the threshold values, I think maybe state what the threshold values are, as it is hard to read them off of Figure 7. Also you state that for stages 1 and 3, the reactivity increases with increasing vibrational excitation. I can see this for stage 1, i.e. that at low collision energies the ICS is higher for $v = 1$ and $v = 2$ than for $v = 0$. However, I'm insure where

stage 3 begins, as in Figure 7 at higher collision energies the ICS for $v = 1$ and $v = 2$ is always lower than for $v = 0$. You also state the threshold tends to decrease for increasing vibrational excitation, I can see this in Figure 7, but again it might be useful to give the threshold energies as they are difficult to read off the Figure.

**Reply:** Thanks for this suggestion. We have rephrased this paragraph, as shown below:

Lines 344 to 364 **-** 'As shown in Fig. 7, both QCT and TDWP ICSs rise rapidly from the threshold in stage 1 and gradually reach a plateau (stage 2), and then increase again and finally stabilize (stage 3). Also, the TDWP ICS at $v = 0$ is larger than the QCT ICS at $v = 0$ due to the quantum effects. The threshold is the minimum collision energy that the reaction can occur (i.e. the point of intersection of ICS and the X-axis), which are about 0.1 for TDWP ICS at $v = 0$ and 0.133, 0.11 and 0.09 eV for QCT ICSs at $v = 0$, $v = 1$ and $v = 2$, respectively. The threshold of QCT ICS at $v = 0$ is consistent with the entrance barrier height but the threshold of TDWP ICS at $v = 0$ is less than it, because the TDWP method includes the tunnelling effect and the zero-point-energy (ZPE) correction.

Furthermore, the vibrational excitation of $O_2$ has different effects on the reactivity for the three stages of this reaction. For the stage 1, the reactivity increases with increasing vibrational excitation, because the increased vibrational energy facilitates the reaction through the path with a barrier, resulting in less collision energy required for high vibrational states. Also, the threshold tends to decrease for the increasing vibrational excitations. For the stage 2, where the exothermicity dominates, the vibrational excitations of the reactants suppress the reaction reactivity. For the stage 3, both MEP and new entrance channel are contributing to the reaction, in which the reaction through MEP is dominated by the exothermicity like stage 2 and the reaction through the new entrance channel is dominated by the entrance barrier (SP2) like stage 1. In other words, the reaction through the new entrance channel is promoted by the vibrational excitation, so the slopes of QCT ICSs at $v = 1$ and $v = 2$ are significantly greater than QCT ICS at $v = 0$, as shown in Fig. 7. Moreover, the vibrational excitations of the reactants suppress the reactivity of the reaction through MEP and the reaction mainly occurs in this channel. Hence, the combined effect of the vibrational excitation for the two channels in stage 3 is that the vibrational excitations of the reactants suppress the reaction reactivity.'

**#Comment 7:** Around line 380 – you state that 'the experimental rate constants include other possible processes with excited states of $O_2$, O, and PO, that can be responsible for the depletion of $P(^4S)$.' In the more

recent study by Douglas et. al. (2019), 248 nm light is used, at which $O_2$ has as cross section of $< 1e^{-24}$ cm$^2$ molecule$^{-1}$, suggesting interference from excited states of $O_2$ or O atoms are unlikely. However, there may be some O atoms and PO molecules produced from the reaction of $P(^2P, {}^2D) + O_2$ present, although these would be present at very low concentrations compared to the $O_2$ co-reagent. I think is it safer to say the experimental results may be suffering from secondary chemistry, rather than stating that they definitely are.

**Reply:** Thanks for this suggestion. We have rephrased this paragraph, as shown below:

Lines 390 to 391 - 'It is partly because the experimental results may be suffering from secondary chemistry, i.e. some O atoms and PO molecules could be produced from the reactions of $P(^2P, {}^2D) + O_2$, albeit at extremely low quantities.'

**#Comment 8:** Line 396 – This sentence suggests that both the Douglas et. al. (2019) results and the Gomes et. al. (2022) results are both fitted from calculated data below 1000 K (when the earlier work is fitted to experimental data).

**Reply:** Thanks for this suggestion. We have rephrased this sentence, as shown below:

Lines 392 to 394 - 'The theoretical rate constants based on the RRKM theory (Douglas et al., 2019) and TST (Gomes et al., 2022) are also shown in Fig. 8. The RRKM rate constant available for temperatures of 150-1400 K was optimized by the experimental data and the TST rate constant was fitted from the calculated values at the temperatures below 1000 K.'

**#Comment 9:** Around line 405 – Here you discuss the barrier heights of TS1 from different levels of theory. You go on to mention that the barrier height used by Gomes et. al. (2022) is reduced from 0.137 eV down to 0.105 eV when including zero-point energy (ZPE), and it is this lower value that is used in the TST study to calculate rate coefficients. I'm not overly familiar with the QCT and TDWP methods for calculating rate coefficients, but I assume you do not need to include ZPE in your calculations as this is already accounted for? You do mention later on that the relatively small QCT rate constants at low temperature are due to the ZPE problem, whereas the TDWP rate constants are faster, with the threshold agreeing well with the 0.105 eV barrier height that includes ZPE. Does this suggest the TDWP rate coefficients take into account ZPE, while the QCT ones do not (and so a ZPE adjustment is required)?

**Reply:** Many thanks for pointing this out. Your understanding is correct that the QCT method is semi-classical and does not consider some quantum effects such as the ZPE problem and tunnelling effect.

The TST method also does not consider the ZPE correction. However, the TST calculation is carried out on a two-dimensional reaction path (i.e. potential energy vs reaction coordinate), so the barrier height can be easily adjusted manually to obtain a reaction path that considers the ZPE correction.

Our dynamic calculations from both QCT and TDWP methods are carried out on a four-dimensional PES (i.e. potential energy vs $R_1$, $R_2$ and $R_3$), which is analytically fitted by thousands of *ab initio* energy points. Hence, it is difficult to manually adjust the barrier on the PES. The QCT and TDWP methods use the same PES with an entrance barrier height of 0.133 eV. The threshold of QCT ICS (of 0.133 eV) is consistent with the entrance barrier height but the threshold of TDWP ICS (of 0.1 eV) is less than them, because the TDWP method from pure quantum mechanics includes the tunnelling effect and the ZPE correction. As expected, the TDWP threshold agrees well with the modified TST barrier height (of 0.105 eV) that includes ZPE correction. The difference of the barrier height (of 0.05 eV) may be due to the tunnelling effect.

We have specifically pointed out it in the discussion about the threshold, as shown below:

Lines 349 to 350 - 'The threshold of QCT ICS at $v = 0$ is consistent with the entrance barrier height but the threshold of TDWP ICS at $v = 0$ is less than them, because the TDWP method includes the tunnelling effect and the zero-point-energy (ZPE) correction.'

Lines 410 to 413 - 'The TDWP rate constants consider the ZPE correction and the tunnelling through the entrance channel barrier and the threshold of ICS is about 0.1 eV, agreeing with the modified barrier height (0.105 eV) used in the TST calculation. The difference of the barrier height (0.05 eV) may be due to the tunnelling effect.'

**#Comment 10:** Around line 425 – your results nicely show that above 3000 K the ro-vibrational excitation of the $O_2$ enhances the rate coefficient. However, your predicted rate coefficients at high temperatures are still lower than those of Gomes et. al. (2022) who also consider the [4]A state in their calculations. Do you need to consider this state in your calculations too, and if not does this suggest your rate coefficients at very high temperatures are underestimated?

**Reply:** Many thanks for pointing this out. Your understanding is correct. Our current rate constants at very

high temperatures are underestimated, because we do not consider the contribution from the quartet state. However, the entrance barrier height including ZPE correction for the quartet state (0.3 eV) is about three times above that of the doublet state (0.105 eV) (Gomes et. al. 2022), so the effect of quartet state on the rate constant may be small.

We guess that you may misunderstand the TST rate constants from Gomes et. al. (2022) in Fig.8. The TST rate constant curve is drawn by the Arrhenius formula [$1.44 \times 10^{-12}(T/300)^{1.66}\exp(-600/T)$] fitted by the calculated values below 1000 K **without high-temperature data backing**. Since the rate constant will deviate from the Arrhenius behaviour at high temperatures, the TST rate constant cannot guarantee its accuracy at high temperatures, especially for temperatures above 2000 K. Therefore, the fitting of the high temperature rate constant formula requires sufficient data support (sometimes it is necessary to divide into several temperature ranges to fit the formula separately.) and the reference value of the TST rate constant at very high temperatures remains to be verified. We can guarantee that our QCT and TDWP results have sufficient data support below 5000 K and the TDWP method is advanced enough.

It is difficult to consider the quartet state in our work, because our treatment for obtaining the rate constant needs huge computational cost, especially for the TDWP calculation of this reaction involving three heavy atoms and deep potential wells. For one 32-core server we use, the CPU time for 6471 *ab initio* energies, QCT (all the ICSs and rate constants in our work) and TDWP (all the reaction probabilities of $J = 0$ to 270, ICS and rate constant in our work) calculations are about 1600, 500 and 11000 h, respectively.

We have rephrased the relevant descriptions, as shown below:

Lines 396 to 400 - 'We did not consider the contribution from the quartet state due to the extremely high computational cost, therefore our rate constants at very high temperatures may be slightly underestimated. However, the entrance barrier height including ZPE correction for the quartet state (0.3 eV) is about three times above that of the doublet state (0.105 eV) (Gomes et al., 2022), so the effect of quartet state on the rate constant may be small.'

Lines 417 to 419 - 'At high temperatures, it is necessary to point out that the previous two theoretical Arrhenius formulas (Douglas et al., 2019; Gomes et al., 2022) were fitted without high-temperature rate constants backing. Since the rate constant will deviate from the Arrhenius behaviour at high temperatures, the RRKM and TST rate constants remains to be verified at high temperatures.'

**#Comment 11:** Line 433 – you give the parameterized rate coefficients in the supplemental, I think it would be good to have these in the main paper for ease of astro/atmospheric modellers requiring these rate coefficients.

**Reply:** Thanks for this suggestion. We have added the parameterized rate constants in the main paper, as shown below:

Lines 431 to 437 - 'The rate constants of TDWP ($v = 0, j = 0$), QCT ($v = 0, j = 0$) and QCT (Thermal) for the P($^4$S) + O$_2$(X$^3\Sigma^-$) → O($^3$P) + PO(X$^2\Pi$) reaction can be approximated using the three-parameter Kooij function, given by (Laidler, 1996)

$$k(T) = A\left(\frac{T}{300}\right)^{\alpha} e^{-\beta/T}$$

(2)

where $A$, $\alpha$, and $\beta$ are fitting parameters. The rate constant curves are divided into four temperature ranges, and the fitting parameters are summarized in **Table 1**. The fitted rate constants deviate less than 1 per cent from our calculated ones. These rate constants are calculated in the atom-diatom system and independent of pressure, which may be applicable to the ISM and upper planetary atmospheres.'

**Table 1 Parameters for Kooij function obtained by fitting the computed rate constants.**

| Method | $T$ / K | $A$ / cm$^3$s$^{-1}$ | $\alpha$ | $\beta$ / K |
|---|---|---|---|---|
| TDWP ($v = 0, j = 0$) | 100 - 600 | $2.2508 \times 10^{-12}$ | 1.3554 | 1041.9940 |
| | 600 - 1000 | $4.4682 \times 10^{-12}$ | 0.9430 | 1283.2565 |
| | 1000 - 2000 | $8.6398 \times 10^{-12}$ | 0.6532 | 1596.5685 |
| | 2000 - 5000 | $6.7105 \times 10^{-11}$ | -0.0146 | 3193.4739 |
| QCT (Thermal) | 200 - 600 | $4.7988 \times 10^{-13}$ | 2.5132 | 734.5248 |
| | 600 - 1000 | $6.7196 \times 10^{-12}$ | 0.7068 | 1574.9928 |
| | 1000 - 2000 | $6.7933 \times 10^{-12}$ | 0.7215 | 1610.7254 |
| | 2000 - 5000 | $2.0656 \times 10^{-12}$ | 1.0844 | 590.6460 |
| QCT ($v = 0, j = 0$) | 2000 - 5000 | $5.7104 \times 10^{-11}$ | 0.0244 | 3237.2285 |

**#Comment 12:** I have another more general comment. The title of your paper includes 'implications for atmospheric modelling', however in the manuscript you don't actually mention what the implications of the new rate coefficient are, say for Earth's atmosphere or the interstellar medium. Thinking of implications for atmospheric modelling, you also don't mention at what pressures your new rate coefficients are applicable. I'm assuming both the QCT and TDWP methods give low pressure limiting (or zero pressure) rate coefficients,

which will be applicable to the ISM and upper planetary atmospheres where the pressure is low?

**Reply:** We thank the reviewers for this important point. We have added the relevant descriptions, as shown below:

Lines 435 to 437 - 'These rate constants are calculated in the atom-diatom system and independent of pressure, which may be applicable to the ISM and upper planetary atmospheres.'

Lines 439 to 463 - '**6 Atmospheric Implications**

The ablation of IDPs in the upper atmosphere (mainly at heights between 70 and 110 km) of terrestrial planets delivers about 6,200 kg $yr^{-1}$ ablated phosphorus to the atmosphere (Carrillo-Sánchez et al., 2020), where the temperatures ranging from below 100 to over 2500 K (in ablation IDPs). Several reaction networks of meteor-ablated phosphorus in the Earth's upper atmosphere (Douglas et al., 2019; Douglas et al., 2020; Plane et al., 2021) indicate that the initial oxidation of P will proceed through the successive oxidation by $O_2$ to produce OPO (i.e. reactions R1 and R2). The oxidation by $O_3$ is not significant because $O_2$ is $10^5$ times more abundant than $O_3$ at this altitude. Also, OPO is likely dissociated into PO and P as a result of hyperthermal collisions with air molecules (Carrillo-Sánchez et al., 2020). Therefore, the $P + O_2 \rightarrow O + PO$ reaction may occur throughout the upper mesosphere and thermosphere. Our rate constants are fitted by sufficient data below 5000 K, which is appropriate for most altitude of the Earth's atmosphere and can be used to model its phosphorus chemistry.

A recent study developed a reaction network of phosphorus atmosphere chemistry (Plane et al., 2021), including the possible routes from $P + O_2 \rightarrow O + PO$ to the stable reservoirs ($H_3PO_3$ and $H_3PO_4$). Subsequently, they incorporated the rate constants of the associated reactions into the Whole Atmosphere Community Climate Model from the US National Center for Atmospheric Research (Gettelman et al., 2019), and then explored the vertical profiles of the P-containing species and the global mean P deposition flux. Also, they estimated that the fraction of the ablated phosphorus forming bioavailable metal phosphites was 11%.

One of our concerns is that the theoretical predicted rate constants of $P + O_2 \rightarrow O + PO$ diverge from experimental result at 200 K, which is about the typical temperature of the upper mesosphere and lower thermosphere. For the reaction with entrance barrier, its rate constant usually has Arrhenius linear behaviour at relative low temperatures. Our QCT and TDWP rate constants, as well as the TST rate constant (Gomes et

al., 2022), exhibit highly consistent Arrhenius linear behaviour at relative low temperatures, with slopes close to those of the experimental rate constants at about 300-600 K (Douglas et al., 2019), as shown in Fig. 8 (b). There is reason to believe that the experimental results of 300-600K are very reliable. Whereas, the experimental rate constant at about 200 K diverges from this linear behaviour and seems to be slightly overestimated, so the lifetime of the P atoms in the atmosphere may be a little longer than previously thought (Carrillo-Sánchez et al., 2020). Further experiments for the rate constants at approximately 200 K of P + $O_2$ → O + PO are encouraged in the future.'

**Technical Corrections:**

**#Correction 1:** Line 33 – change planet to planets.

**Reply:** Thanks for this suggestion. We have changed 'planet' to 'planets' in line 33, as follows:

Line 32 - 'The ablation of IDPs in the upper atmosphere of terrestrial planets delivers PO and P…'

**#Correction 2:** Line 46 – change container to reservoir.

**Reply:** Thanks for this suggestion. We have changed 'container' to 'reservoir' in line 46, as follows:

Line 44 - '… considered to be the main reservoir for gas-phase P in the ISM …'

**#Correction 3:** Line 49 – add 'the' – 'in the ISM'.

**Reply:** Thanks for this suggestion. We have changed 'in ISM' to 'in the ISM' in line 49, as follows:

Line 47 - 'the investigation on the formation of PO is helpful for modelling its abundance in the ISM.'

**#Correction 4:** Line 55 – strictly speaking, the rate coefficients for P($^2$D, $^2$P) + $O_2$ are only loss rates, as the branching ratio between chemical reaction forming O + PO, and electronic relaxation forming P($^4$S) + $O_2$ is not known.

**Reply:** Thanks for this suggestion. We have removed the '$^2$D, $^2$P' this sentence, as shown below:

Lines 52 to 53 - 'A recent experiment has determined the rate constants of P ($^4$S) + $O_2$ → O + PO at temperatures

ranging from ~200 to 750 K (Douglas et al., 2019).'

**#Correction 5:** Line 78 / 79 – add 'the' – 'that the potential energy surface'.

**Reply:** Thanks for this suggestion. We have changed 'that potential energy surface' to 'that the potential energy surface' in lines 78 to 79, as follows:

Lines 73 to 74 - 'It is worth noting that the potential energy surface (PES) can yield …'

**#Correction 6:** Line 81 – add 'of' – 'such as the reaction or non-reaction of collisions, and'.

**Reply:** Thanks for this suggestion. We have changed 'the reaction or non-reaction collisions' to 'the reaction or non-reaction of collisions' in line 81, as follows:

Line 76 - 'such as the reactive or non-reactive of collisions and photodissociation within the system.'

**#Correction 7:** Line 169 – remove 'the' (to construct a global PES of $PO_2$).

**Reply:** Thanks for this suggestion. We have removed 'the' in line 169, as follows:

Line 161 - '… to construct a global PES of $PO_2$'

**#Correction 8:** Line 239 – change display to displays.

**Reply:** Thanks for this suggestion. We have changed 'display' to 'displays' in line 239, as follows:

Lines 224 to 225 - 'which displays the smooth long-range behavior of the CHIPR PES.'

**#Correction 9:** Line 241 – rephrase sentence – e.g. change to 'When the Jacobi approaching angle is about 40 - 50°, it is much easier for the P atom to cross the barrier (TS1) and reach the LM'.

**Reply:** Thanks for this suggestion. We have rephrased this sentence in line 241, as follows:

Lines 226 to 227 - 'When the Jacobi approaching angle is about 40 ~ 50°, it is much easier for the P atom to cross the barrier (TS1) and reach to the LM, ...'

**#Correction 10:** Line 246 – maybe change evolved to reached – and add the – 'both TS3 and TS4 can be reached from the LM'.

**Reply:** Thanks for this suggestion. We have changed 'evolved' to 'reached' and added 'the' in line 246, as follows:

Line 231 - '…, both TS3 and TS4 can be reached from the LM, ...'

**#Correction 11:** Line 261 – change possible to able, change to (at end of sentence) to the – 'At high collision energies, the P atom is able to cross the $C_{2V}$ barrier SP2 and reach the GM directly, …'.

**Reply:** Thanks for this suggestion. We have changed 'possible' to 'able' and changed 'to' to 'the' in line 261, as follows:

Lines 246 to 247 - 'At high collision energies, the P atom is able to cross the $C_{2v}$ barrier SP2 and reach the GM directly, …'

**#Correction 12:** Line 280, equation 9 – change P + H$_2$ to P + O$_2$.

**Reply:** Thanks for this suggestion. We have changed 'P + H$_2$' to 'P + O$_2$' in equation 9, as follows:

$$k(T) = g_e(T) \left( \frac{8k_B T}{\pi \mu_{P+O_2}} \right)^{1/2} \pi b_{max}^2 \frac{N_r}{N} \tag{3}$$

**#Correction 13:** Line 298 – add 'the' – 'is the Jacobi form of the CHIPR PES'.

**Reply:** Thanks for this suggestion. We have added 'the' in line 298, as follows:

Line 286 - '… $V(R, r)$ is the Jacobi form of the CHIPR PES for PO$_2$ …'

**#Correction 14:** Line 337 – descending.

**Reply:** Thanks for this suggestion. We have amended 'dencending' to 'descending' in line 337, as follows:

Line 324 - 'The second stage is the descending one after the peak, ...'

**#Correction 15:** Line 344 – add which – 'bound and quasi-bound states which exist in the LM and GM potential wells'.

**Reply:** Thanks for this suggestion. We have added 'which' in line 344, as follows:

Lines 335 to 336 - '… bound and quasi-bound states which exist in the LM and GM potential wells, ...'

**#Correction 16:** Line 347 – remove 'a' – 'resulting in plenty of sharp'.

**Reply:** Thanks for this suggestion. We have rewritten this sentence, as follows:

Lines 338 to 339 - 'Hence, the bondage of potential well is strong at low collision energy, resulting in the long lifetimes of the collision complexes and the plenty of sharp and violent resonances before the peak (stage 1).'

**#Correction 17:** Line 406 – change 'to' to 'in' – 'and then used in the TST study'.

**Reply:** Thanks for this suggestion. We have changed 'to' to 'in' in line 406, as follows:

Lines 405 to 406 - '… and then used in the TST study.'

**#Correction 18:** Line 417 – add the – 'At high temperatures, the previous two …'.

**Reply:** Thanks for this suggestion. We have rewritten this sentence, as follows:

Lines 417 to 418 - 'At high temperatures, it is necessary to point out that the previous two theoretical Arrhenius formulas (Douglas et al., 2019; Gomes et al., 2022) were fitted without high-temperature rate constants backing.'

**#Correction 19:** Line 444 – remove s from channels and change to reasonably – 'The long-range interactions, diatomic potentials, and dissociation energies of each asymptotic channel are reasonably reproduced.'.

**Reply:** Thanks for this suggestion. We have removed 's' from 'channels' and changed 'reasonable' to 'reasonably' in line 444, as follows:

Lines 468 to 469 - 'The long-range interactions, diatomic potentials and dissociation energies of each asymptotic channel are reasonably reproduced.'

**#Correction 20:** Line 458 – remove 'the', and maybe change reaction to reactive? – 'can be used for molecular simulations of reactive or non-reactive collisions and photodissociation of the $PO_2$ system in atmospheres and the interstellar medium.'.

**Reply:** Thanks for this suggestion. We have removed 'the' and changed 'reaction' to 'reactive' in line 458, as follows:

Lines 481 to 482 - 'The presented CHIPR PES of $PO_2(X\,^2A_1)$ can be used for molecular simulations of reactive or non-reactive collisions and photodissociation of the $PO_2$ system in atmospheres and interstellar medium.'

---

## Author Comment (AC2)

**Response to the reviewer 2**

In this work, Chen, Qin et al. performs a systematic and exhaustive quantum chemistry study on the reaction of the collision reaction between P and molecular oxygen giving rise to O and PO with relevance in the Earth's atmospheric chemistry of phosphorus and its interstellar chemistry. While previous theoretical studies focused on providing "static information", in this case, the authors performed both semi-classical and quantum dynamics simulations to provide with time-dependent properties. The quality of the study is high and the methods and techniques are the most state-of-the-art ones for this type of problems. The presentation of the manuscript is also very clear, systematic and precise.

**Reply:** We thank the reviewer for their comments which we address below.

**Specific Comments:**

**#Comment 1:** Even though in the title is it stated "… implication for atmospheric modelling", there is no much discussion on how the new data obtained from the dynamics simulations would impact the chemistry in the atmosphere and interstellar media of the studied system. More details and illustrative examples would help to enhance the scientific significance of the work.

**Reply:** We thank the reviewers for this important point. We have added the relevant descriptions, as shown below:

Lines 439 to 463 - '**6 Atmospheric Implications**

The ablation of IDPs in the upper atmosphere (mainly at heights between 70 and 110 km) of terrestrial planets delivers about 6,200 kg $yr^{-1}$ ablated phosphorus to the atmosphere (Carrillo-Sánchez et al., 2020), where the temperatures ranging from below 100 to over 2500 K (in ablation IDPs). Several reaction networks of meteor-ablated phosphorus in the Earth's upper atmosphere (Douglas et al., 2019; Douglas et al., 2020; Plane et al., 2021) indicate that the initial oxidation of P will proceed through the successive oxidation by $O_2$ to produce OPO (i.e. reactions R1 and R2). The oxidation by $O_3$ is not significant because $O_2$ is $10^5$ times more abundant than $O_3$ at this altitude. Also, OPO is likely dissociated into PO and P as a result of hyperthermal collisions with air molecules (Carrillo-Sánchez et al., 2020). Therefore, the P + $O_2$ → O + PO reaction may occur throughout the upper mesosphere and thermosphere. Our rate constants are fitted by sufficient data below 5000

K, which is appropriate for most altitude of the Earth's atmosphere and can be used to model its phosphorus chemistry.

A recent study developed a reaction network of phosphorus atmosphere chemistry (Plane et al., 2021), including the possible routes from $P + O_2 \rightarrow O + PO$ to the stable reservoirs ($H_3PO_3$ and $H_3PO_4$). Subsequently, they incorporated the rate constants of the associated reactions into the Whole Atmosphere Community Climate Model from the US National Center for Atmospheric Research (Gettelman et al., 2019), and then explored the vertical profiles of the P-containing species and the global mean P deposition flux. Also, they estimated that the fraction of the ablated phosphorus forming bioavailable metal phosphites was 11%.

One of our concerns is that the theoretical predicted rate constants of $P + O_2 \rightarrow O + PO$ diverge from experimental result at 200 K, which is about the typical temperature of the upper mesosphere and lower thermosphere. For the reaction with entrance barrier, its rate constant usually has Arrhenius linear behaviour at relative low temperatures. Our QCT and TDWP rate constants, as well as the TST rate constant (Gomes et al., 2022), exhibit highly consistent Arrhenius linear behaviour at relative low temperatures, with slopes close to those of the experimental rate constants at about 300-600 K (Douglas et al., 2019), as shown in Fig. 8 (b). There is reason to believe that the experimental results of 300-600K are very reliable. Whereas, the experimental rate constant at about 200 K seems to be slightly overestimated, so the lifetime of the P atoms in the atmosphere may be a little longer than previously thought (Carrillo-Sánchez et al., 2020). Further experiments for the rate constants at approximately 200 K of $P + O_2 \rightarrow O + PO$ are encouraged in the future.'

**Technical Corrections:**

**#Comment 1:** On page 4, equation (1), define R1, R2 and R3.

**Reply:** Thanks for your advice. We have redefined this equation, as follows:

$$PO_2(X\,^2A_1 / ^2A') \rightarrow O(^3P) + PO(X\,^2\Pi) \qquad\qquad \text{R1}$$
$$\rightarrow P(^4S) + O_2(X\,^3\Sigma^-) \qquad\qquad \text{R2}$$
$$\rightarrow P(^4S) + O(^3P) + O(^3P) \qquad\qquad \text{R3}$$

**#Comment 2:** On page 6, line 169, change "…to construct a global the PES…" by "…to construct a global PES…"

**Reply:** Thanks for your advice. We have removed 'the' in line 169, as follows:

Line 161 - '… to construct a global PES of $PO_2$'

**#Comment 3:** In relation to Table 2, please define more clearly the meaning of "ascending ordered energies" and the meaning of the numbers.

**Reply:** Thanks for your advice. We have rewritten the definition, as follows:

Lines 173 to 175 - 'The final PES was constructed from 6471 *ab initio* energy points, covering an energy range up to 500 kcal/mol above the GM. Table 2 lists the RMSDs between the analytical CHIPR energies and *ab initio* energies in the indicated energy range above the GM and those for the additional energy grids.'

**Table 2 The root-mean-square deviations (RMSDs) in the indicated energy range above the GM and those for the additional energy grids.**

| | $N$ [a] | Max deviation [b] | RMSD [c] | $N_{>\text{RMSD}}$ [d] |
|---|---|---|---|---|
| Energy Range [e] | | | | |
| 10 | 1560 | 95.4 | 24.6 | 447 |
| 20 | 1892 | 103.9 | 23.1 | 504 |
| 40 | 1986 | 152.0 | 24.4 | 485 |
| 60 | 2039 | 185.5 | 29.4 | 439 |
| 80 | 2131 | 226.5 | 37.9 | 339 |
| 100 | 2231 | 263.9 | 52.9 | 253 |
| 200 | 5911 | 280.3 | 84.4 | 1218 |
| 300 | 6415 | 319.1 | 90.1 | 1441 |
| 500 | 6471 | 387.9 | 91.5 | 1461 |
| Configuration [f] | | | | |
| GM | 1554 | 79.7 | 24.5 | 452 |
| LM | 490 | 71.4 | 21.4 | 140 |
| TS1 | 810 | 48.6 | 20.3 | 292 |
| TS2 | 405 | 29.8 | 10.5 | 118 |
| TS3 | 810 | 34.1 | 13.4 | 242 |
| TS4 | 810 | 60.8 | 14.5 | 188 |

[a] The number of energy points in the corresponding range. [b] The maximum deviation in the corresponding range, $cm^{-1}$. [c] The RMSD for the corresponding range, $cm^{-1}$. [d] The number of energy points with a deviation larger than the RMSD. [e] The indicated energy range above the GM, kcal $mol^{-1}$. [f] The additional energy grids.

**#Comment 4:** In Fig. 2, it would help the reader if $E_h$ is defined.

**Reply:** Thanks for your advice. Since the $E_h$ is first used in the Fig. 1, we have added the corresponding definition on Fig. 1 caption, as follows:

Line 151 – 'Fig. 1 (a) The PECs of $O_2(X\ ^3\Sigma^-)$ and $PO(X\ ^2\Pi)$. The unit of potential energy is the atomic unit (Hartree, $E_h$).'

**#Comment 5:** On page 8, line 210, change "Fig. 4(a)" by "Fig. 2(a)".

**Reply:** Thanks for your advice. We have changed 'Fig. 4(a)' by 'Fig. 2(a)' in line 210 as follows:

Line 199 - 'Fig. 2 (a) presents the channel of an O atom dissociated from $PO_2(X\ ^2A_1)$ …'

**#Comment 6:** On page 8, line 211, rewrite this sentence since the O atom dissociation indeed requires energy and therefore it is not "barrierless".

**Reply:** Thanks for your advice. We have removed 'barrierless' in line 211, as follows:

Line 199 - 'Fig. 2 (a) presents the channel of an O atom dissociated from $PO_2(X\ ^2A_1)$ …'

**#Comment 7:** On Fig. 4 caption, add at the end of the first sentence "(see definition in the text)" referring to the hyperspherical coordinates.

**Reply:** Thanks for your advice. We have added '(see definition in the text)' on Fig. 4 caption, as follows:

Line 258 - 'Fig. 2 The relaxed triangular contour plot for the ground-state $PO_2$ in hyperspherical coordinates (see the definition in the text).'

**#Comment 8:** In the last paragraph of Conclusions, re-formulate the grammar of the sentence "It can also be a reliable component for constructing other larger molecular systems containing $PO_2$, such as $PO_3$ and $HPO_2$ correspond to the reactions R2 and R3 for generating $H_3PO_3$ in the Earth's atmosphere."

**Reply:** Thanks for your advice. We have re-formulated the grammar of this sentence, as follows:

Lines 482 to 484 - 'Moreover, this analytical PES of $PO_2$ can provide reliable two-body fragments and three-body fragment for the construction of $PO_3$ and $HPO_2$ PESs using MBE form, so as to carry out dynamic study of reactions R2 and R3, which are also important for generating $H_3PO_3$ in the Earth's atmosphere.'

**#Comment 9:** It would also help the reader if Jacobian coordinated are briefly defined.

**Reply:** Thanks for your advice. We have added the definition of Jacobian coordinate, as follows:

Lines 104 to 105 – 'For example, the $R_{A\text{-}BC}$, $r_{BC}$ and $\gamma_{A\text{-}BC}$ for the A-BC channel of ABC molecular are defined as the distance of the A atom relative to the center-of-mass of BC, the bond distance of BC and the angle between $R_{A\text{-}BC}$ and $r_{BC}$, respectively.'

---

## Author Response (AR2)

**Response to the reviewer 2**

Thanks very much for your advice.

**Technical Corrections:**

**#Comment 1:** Please change "…are very reliable. Whereas, the experimental rate constant at about 200 K diverges…" by "…are very reliable, whereas the experimental rate constant at about 200 K diverges…".

**Reply:** Thanks for this suggestion. We have rephrased this sentence, as follows:

Lines 260 to 263 - 'There is reason to believe that the experimental results of 300-600K are very reliable, whereas the experimental rate constant at about 200 K diverges from this linear behaviour and seems to be slightly overestimated. Therefore, the lifetime of the P atoms in the atmosphere may be a little longer than previously thought (Carrillo-Sánchez et al., 2020).'